# Socio-ecological impacts of the 2025 Los Angeles urban fires on communities, neighborhoods, and homes

Carl A. Norlen ®[1,2] ✉, Sadikshya Sharma[1] & Francisco J. Escobedo ®[1]

Human settlements are increasingly being impacted by urban fires initiated by wildfires. Metrics such as area burned and number of structures destroyed are important, but research often overlooks the socio-ecological complexity of urban fires. We study the impacts of the 2025 Los Angeles fires on two communities at the neighborhood and residential parcel scales. Geospatial analyses and econometric modeling explore the relationships between urban morphology, socio-demographic factors, and home destruction. Here we show that socio-ecological characteristics and scale are key in parsing the dynamics of urban fires. Also, new socio-demographic populations are being affected and urban morphology metrics are more important than vegetation cover. Despite parallels with 19th and early 20th century urban conflagrations, understanding these re-emerging urban fires requires transdisciplinary approaches and unique metrics. Investigating the socio-ecological scales and dynamics of urban fires provides a valuable next step towards understanding and adapting to the risk associated with these disasters.

There is an emerging trend in which urbanized human settlements are increasingly impacted by extreme and costly fire events[1,2]. Such wildfire-driven events in Wildland-Urban Interface (WUI) zones regularly occur in places such as western North America, Australia, and southern Europe[3–7]. However, these fire events are now occurring frequently in other highly populated urban areas, and across different biomes[1]. The social, economic, and environmental costs are increasingly becoming more catastrophic as experienced in Paradise, California, in 2018[8], 2023 in Lahaina, Hawaiʻi, and in the 2024 Valparaiso region fires in Chile[9]. The recent 2025 Los Angeles (L.A.) California wildfires affected highly urbanized, socio-demographically diverse communities, and the impacts and costs have been estimated at $76 to 131 billion US Dollars (USD)[10]. Such urban conflagrations were historically common in cities prior to the 1900s or during periods of armed conflict[11]. Accordingly, the 2025 L.A. Fires are a harbinger of this re-emerging threat and present an opportunity to better understand these fires. Specifically, the multi-spatial impacts of fires on communities, neighborhoods, homes, and families, as well as urban ecologies,

socio-environmental and economic costs, land use decisions, governance structures, and infrastructures.

The wildfire and WUI literature frequently discuss fires that impact human settlements, or communities hereafter, as events driven by either extreme climate conditions, vegetation biomass or fuels[12–15], other landscape factors, and ignitions[16–18]. Further, these events are often communicated in terms of hectares lost and the number of structures destroyed and evacuees[19]. However, key to understanding impacts from these events is how communities are defined since this has implications for urban ecologies, wildfire risk management, and identifying vulnerable at-risk communities. For example, the term communities in the wildfire literature is a commonly used concept and, to our knowledge, is generally defined based on the number of building footprint clusters across the landscape[20,21]. But, this definition has little consideration for their urban morphology (e.g., pre-fire tree cover, building density), building types (homes versus non-residential structures), human populations (socio-demographic characteristics) and housing stock and tenure[1]. Indeed, available WUI fire risk and

---

[1]U.S. Department of Agriculture, Forest Service, Pacific Southwest Research Station, Riverside, California, USA. [2]U.S. Geological Survey, National Land Imaging Program, Reston, Virginia, USA. ✉e-mail: canorlen@gmail.com

typology literature focuses on wildfire-related factors (e.g., wind speeds, topography) structure density and proximity to wildland vegetation and fuel types. As such, it is often insufficient in capturing the complexity of these disasters in highly urbanized fire-prone communities[4,7,22].

Events like the 2025 Eaton and Palisades fires in L.A. can be considered urban fires initiated by wildfires[23]. Although these types of events are considered unusual in the wildfire science literature[23], these urban conflagrations historically occurred in cities such as Chicago and San Francisco in the late 1800s to early 1900s and intermittently in Europe[24]. While these past urban conflagrations were driven by extreme weather events[25], similar to wildland fires, characteristics of the urban environment (e.g., fences, wooden structures, external features)[26] and associated embers, radiative heat, and fuel loads explain patterns of fire spread and intensity. More recently, these urban fires highlight the importance of urban fuels (e.g., building types, construction materials, and ornamental vegetation)[27] in influencing fire behavior as opposed to wildland fires that are driven by natural vegetation. Urban fires also result in greater loss of life, infrastructure and property as well as indirect impacts such as human displacement and health effects in distant populations due to air, water and soil pollutants (e.g., Lithium from auto batteries, toxic compounds in water systems, soil lead pollution)[28,29].

These disasters represent urban fire events characterized by their severity, rapid spread, and significant impacts on life and property, social structures, urban landscapes, and economic systems[26]. Historically, urban fires have been agents of change and have accelerated existing trends in urban development, such as increased building size and construction types especially in high real estate premium, urban core areas and during economic booms[24]. Their behavior is also different than wildland fires since, rather than being topography or vegetation-driven, they are driven by radiant heat from burning buildings, and ember transport more so than flame contact from vegetation[27]. Less studied is how these fires affect urban communities using a social-ecological systems lens to account for impacts beyond the commonly reported metrics and landscape-scale effects of these events[30,31].

Here, we show that a more nuanced understanding of the complex socio-ecological scales and characteristics of communities impacted by urban fires is needed[32]. Key to this is who lived in these communities and what their exposure, adaptive capacity, and vulnerabilities to urban fires are[30] as well as what types of homes, neighborhoods, and environmental amenities were lost or damaged, as these factors and scales are rarely addressed in the above literature in an integrated manner[30,33]. Scale matters both in analyses of community-specific contexts, fire vulnerability, and potential impacts, as they are highly relevant in mitigating and responding to these events[32]. Accordingly, in this study, we account for scale-dependent differences in the relationships between socio-ecological factors and fire impacts (e.g., modifiable area unit problem, ecological fallacy). This nuanced approach and research will allow society to more effectively mitigate and adapt to these various impacts on communities at risk of urban fires initiated by wildfires[23]. Therefore, our study aims to characterize the social-ecological impacts of the Eaton and Palisades fires (2025 L.A. Fires) at three scales: communities, neighborhoods, and homes. Specifically, our study objectives are to: 1) Compare which socio-ecological factors were associated with home destruction across both communities. 2) Assess how the relationship of socio-ecological factors with home destruction varied depending on the spatial scale of analysis (neighborhood versus parcel-scale).

## Results

### Study area

The study area encompasses the urban communities, neighborhoods, and homes affected by the Eaton and Palisades fires, which occurred in L.A., California, during January 2025 (Fig. 1a). The region's recent climate history, marked by wet winters in 2022–2023 and 2023–24 (double the 1877–2024 mean) followed by hot and dry conditions from April 2024-January 2025 coupled with an extreme wind event on January 7th and 8th[34], created ideal conditions for fire ignition and spread[35]. The Santa Ana winds, with gusts reaching speeds of up to 110 km per hour, exacerbated fire behavior and spread into densely populated urban zones[36]. It is estimated that both fires burned nearly 16,000 homes, businesses, other buildings, and critical infrastructure[37,38], with estimated economic losses reaching $76 to 131 billion USD[10].

These fires primarily impacted urban areas and communities, including the Altadena, Pasadena, and Sierra Madre area (Eaton hereafter); Pacific Palisades (Palisades hereafter); and adjacent wildland areas in the San Gabriel and Santa Monica Mountains. The fires displaced over 180,000 residents and resulted in at least 30 fatalities, underscoring the vulnerability of urban populations to fire events[1,30,39]. The affected areas include social and demographically diverse communities with varying cultural, ethnic and economic status. Palisades is racially homogeneous (80% Non-Hispanic White, 4% Hispanic, 6% Asian/Pacific Islander)[40] and more similar to communities impacted by WUI fires in the past[1], while Eaton is racially diverse (37% Non-Hispanic White, 20% Black/African American, 30% Hispanic, 5% Asian/Pacific Islander)[40]. Overall, Palisade's median per capita income ($140,933) is more affluent than Eaton's ($70,253 median per capita income). In addition, both communities are primarily dominated by single-family residential housing, with Eaton having almost exclusively single-family residences affected, while some areas in Palisades have multi-family residential and mixed residential and commercial land use (Supplementary Fig. 1).

### Community and neighborhood-scale impacts

In 1938 as much as 39% of the community of Palisades was exposed to a fire, and in 1978 and 1993 19% and 21% of its area was exposed to fire, respectively (Fig. 1b). Most of the less densely built neighborhoods in Palisades had been exposed to a fire within the last 100 years, while most of the denser, more urbanized neighborhoods in the southwest corner had not been exposed to a fire before 2025 (Fig. 2). Overall, many of the Eaton community's neighborhoods had no record of fire exposure from 1910–2024 especially in the western, more urbanized neighborhoods and in areas farthest from wildland vegetation (Fig. 2). In contrast, due to the 2025 fires 57% of the Palisades and 53% of the Eaton communities were exposed (Fig. 1b).

Figure 3 displays some representative urban morphology[4,8,9,41] and socio-demographic[1,39,42,43] variables that are influential correlates of home damage in published literature on urban and WUI fires. Additionally, we propose 'structure footprint area' and 'number of structures in Defensible Space Buffer (DSB) Zone 0' as complementary metrics for better understanding urban fires[44,45]. Structure footprint area is a metric that we propose to measure not only density of structures (i.e., number of structures per unit area) as is done in wildfire exposure and WUI studies[7,21], but also to differentiate the size and space these structures occupy per unit area as is done in the urban planning and geography literature[46,47]. Neighborhood-scale analyses show that five out of seven urban morphology variables (Fig. 3a, c; Supplementary Fig. 5 and Fig. 4) produced similar relationships for both Eaton and Palisades, while Urban Tree Cover (UTC) produced divergent relationships (Figs. 3b, 4). In contrast, six out of the fourteen neighborhood-scale socio-demographic variables showed divergent relationships (Figs. 3d–f, 4; Supplementary Fig. 5). For both Eaton and Palisades, neighborhoods with more recent median home construction dates, greater exposure to fire from 1910–2024 (within fire perimeter), and larger median replacement values experienced lower rates of destruction, while neighborhoods with greater structure footprint area experience higher rates of destruction (Figs. 3a, 4). For Palisades, neighborhoods with greater UTC experienced lower rates of

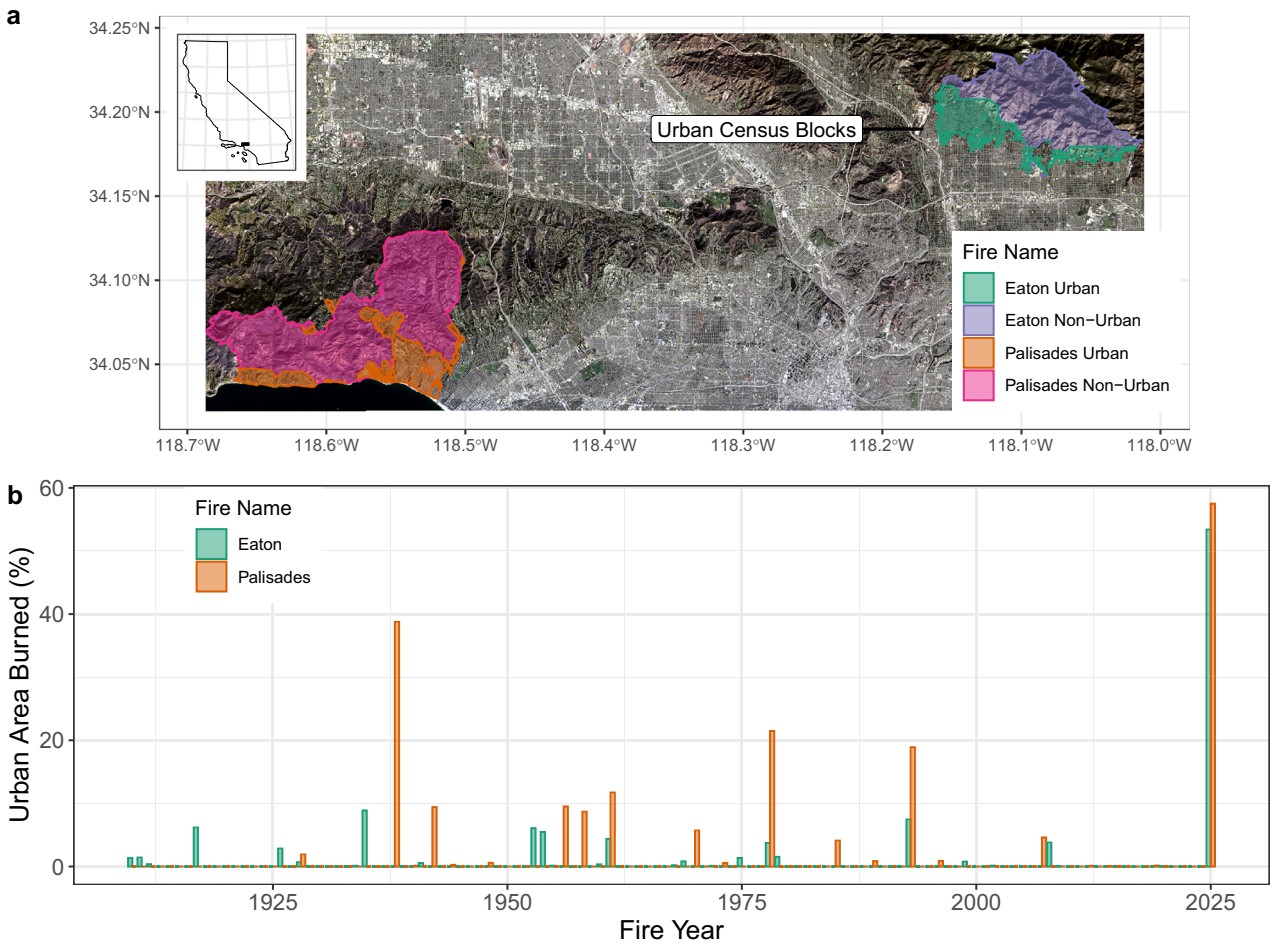

**Fig. 1 | Overview of the study region for the 2025 L.A. fires and history of fire exposure in the Eaton and Palisades communities from 1910-2024.** Panel **a** shows the overlap between Eaton (green) and Palisades (orange) fire perimeters and urban U.S. census blocks or neighborhoods (gray), while panel **b** shows the proportion of the urban neighborhoods exposed to the 2025 Eaton and Palisades fires that were also exposed to fire from 1910–2024. The background image for panel (**a**) is a true color Landsat 9 scene from 1/14/2025, cropped to fit the area of interest.

destruction, while for Eaton, there was a weak but opposite relationship (Figs. 3b, 4). For Palisades, there was a correlation between the number of structures in DSB Zones 0 and 1 and higher rates of destroyed homes, while for Eaton, there was no significant relationship (Fig. 4; Supplementary Fig. 5e).

Supplementary Table 3 shows how pre-fire, neighborhood-scale UTC was slightly higher in Eaton (29.6%) than in Palisades (23.9%). In terms of socio-demographics, Palisades had more elderly and White populations, while Eaton had greater racial and ethnic diversity, including higher Hispanic and African American populations (Supplementary Table 3). Palisades had a greater proportion of 1910–2024 fire exposure (46.9% versus 17.8%), while rates of home destruction were higher in Eaton (69.7%) compared to Palisades (59.7%). Homes in Palisades were newer (median year built = 1963) than in Eaton (median year built = 1946). A greater portion of homes destroyed in Palisades were more recently built (post-2008), and in general had higher median structure replacement values and greater structure footprint area. Homes destroyed in Eaton, on the other hand, had slightly more structures within all DSB Zones, particularly in Zone 2.

**Modeled neighborhood-scale impacts in Eaton**
For Eaton, our neighborhood-scale model (Table 1) shows that 1910–2024 fire exposure was negatively associated with home destruction ($p < 0.001$), while pre-fire UTC showed a positive relationship with destruction ($p < 0.001$). Proportion of residents over 65

years was not significantly associated with destruction, while the proportions of people with an Associate's degree, Bachelor's degree, or below poverty were positively associated with destruction ($p < 0.01$, $p < 0.001$, $p < 0.05$). Similarly, African American populations were positively associated with destruction ($p = 0.001$), while non-English speakers were negatively correlated ($p < 0.001$). We also found that vacant and rented homes, as well as homes with other structures in DSB Zone 0, were not significantly related to destruction. However, home replacement value, proportion of homes built after 2008, and per capita income ($) were negatively associated with destruction ($p < 0.001$, $p < 0.001$, $p < 0.05$). In contrast, structure footprint area was positively associated with home destruction ($p < 0.05$), while the median year homes were built was not significant (Table 1).

**Modeled neighborhood-scale impacts in Palisades**
In Palisades, neighborhood-scale analyses (Table 2) show that pre-fire UTC ($p < 0.1$ in model 2) and 1910–2024 fire exposure were significantly, but marginally, negatively associated with destruction ($p < 0.1$ in model 2). Median year home built was also significantly negatively associated with destruction ($p < 0.01$ in both models). Neighborhoods with more homes built after 2008 had marginally significantly more homes destroyed ($p < 0.1$ in model 2). Similarly, neighborhoods with more structures in DSB Zone 0 were positively associated with destruction ($p < 0.05$, $p < 0.01$), while neighborhoods with greater structure footprint area were not significantly associated

**a**    Palisades

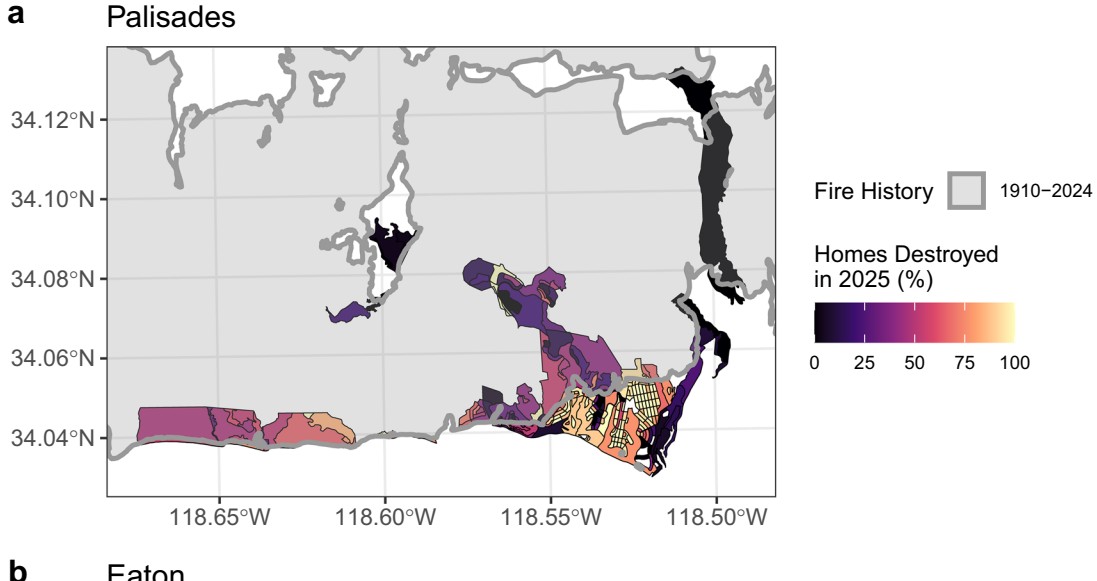

**b**    Eaton

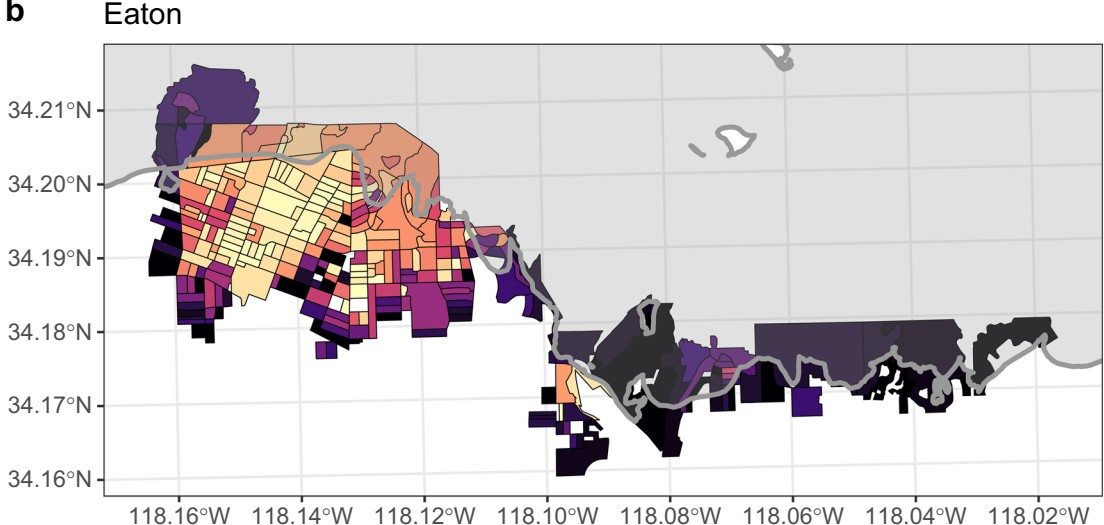

**Fig. 2 | Neighborhood-scale fire impacts for the 2025 L.A. fires.** Fire impacts are measured as % home destruction in each U.S. census block shown with a color scale from black (0%) to yellow (100%) with purple as the midpoint (50%). Impacts are shown for both the **a** Palisades and the **b** Eaton communities, overlayed with polygons (black) for urban 2020 U.S. census blocks impacted by fire in 2025 and fire history from 1910–2024 (gray).

with destruction. Also, White (%), renter (%), and vacant (%) were not significantly associated with destruction, while Hispanic (%) was positively associated with destruction ($p < 0.05$ in model 1, $p < 0.1$ in model 2). Similarly, neighborhoods with more people 65 years and over or 20–64 years old were either not significantly associated or marginally negatively associated with destruction. Higher destruction was also associated with neighborhoods that had greater proportions of people with an Associate's degree or high school level education ($p < 0.01$, $p < 0.1$), while higher per capita income was associated with reduced home destruction ($p < 0.1$ model 1, $p < 0.05$ model 2).

**Parcel-scale impacts in Eaton and Palisades**
At the parcel-scale, homes in Eaton were older (1946) and had lower home replacement values ($280,077) than those in the Palisades (1961; $442,052; Supplementary Table 4), while UTC was greater in Eaton than in Palisades (Supplementary Table 4). Eaton homes showed slightly greater numbers of structures in DSB Zones 0, 1, and 2 and greater pre-fire UTC (29 %), while Palisades had larger structure footprint areas (3121 m² ha⁻¹) and slightly more occupants (2.87 per home) and elderly people (0.13 per home; Supplementary Table 4). In Eaton,

only 19% of homes had been exposed to fire from 1910–2024 compared to 51% of homes in Palisades. Additionally, 86% of Eaton properties were classified as single-family homes, slightly higher than the 79% observed in Palisades. Parcel-scale predicted property owner race based on owner names (see Methods for details) was predominantly non-White, with 28% identified as White in Eaton and 39% in Palisades (Supplementary Table 4). Predicted property owner race results are in line with U.S. Census data for Eaton, however results for Palisades were not consistent (Supplementary Fig. 2), so predicted property owner race data were not used in subsequent analyses due to concerns about accuracy.

Overall, our Eaton parcel-scale model (Table 3) found that pre-fire UTC was significantly and positively associated with destruction ($p < 0.001$) while 1910–2024 fire exposure was strongly and negatively associated with destruction ($p < 0.001$). Both the year home was built, and home replacement value were significantly and negatively associated with destruction ($p = 0.006$, $p < 0.001$). Structure footprint area (m² ha⁻¹) was also strongly negatively associated with destruction ($p < 0.001$), and single residence homes were significantly less likely to be destroyed ($p < 0.001$). Similarly, the number of structures in DSB

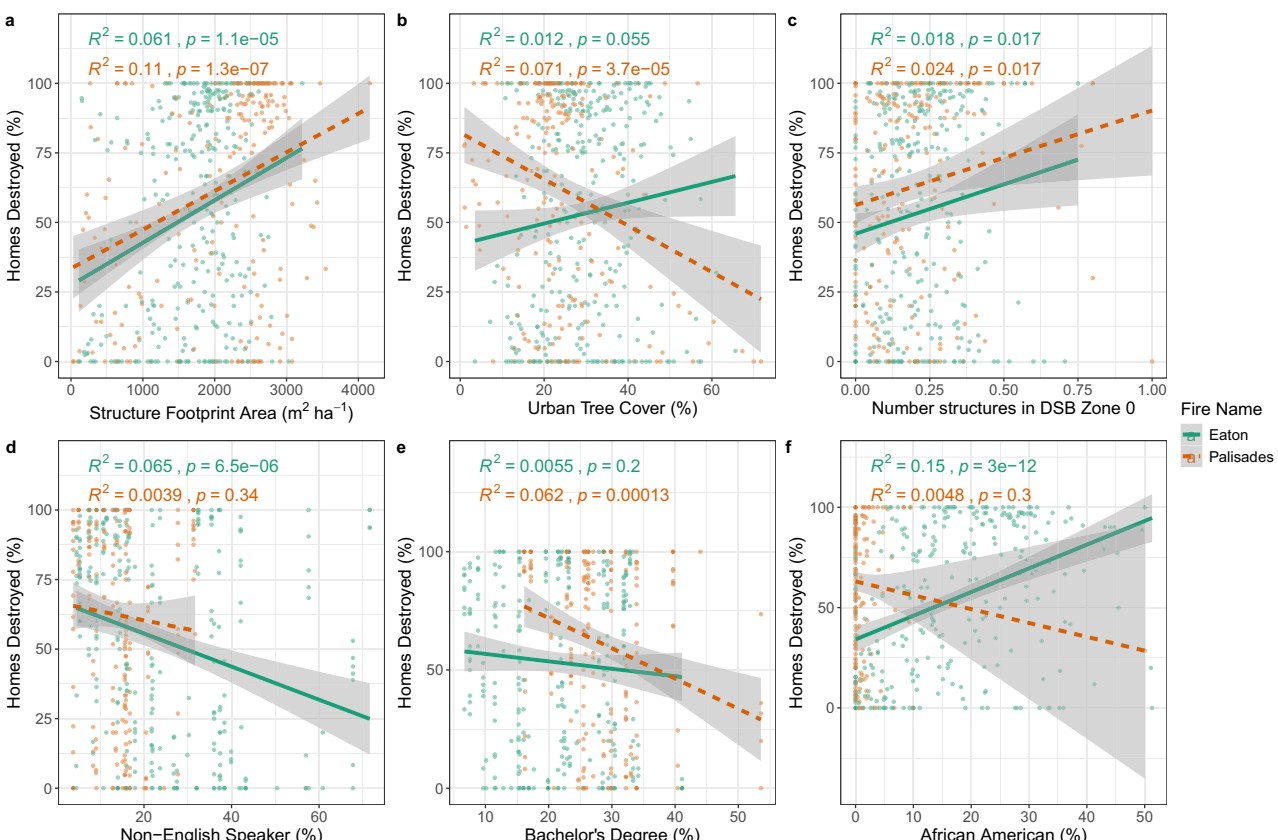

**Fig. 3 | Relationships between neighborhood-scale urban morphology and socio-demographic characteristics and fire impacts.** Each panel shows the linear relationships between key neighborhood-scale urban morphology (**a**–**c**) or socio-demographic characteristics (**d**–**f**) and direct fire impacts (% homes destroyed) for both the Eaton (green) and Palisades (orange) communities. The lines represent ordinary least squares linear regression fits, with the shaded area representing standard errors. DSB = Defensible Space Buffer.

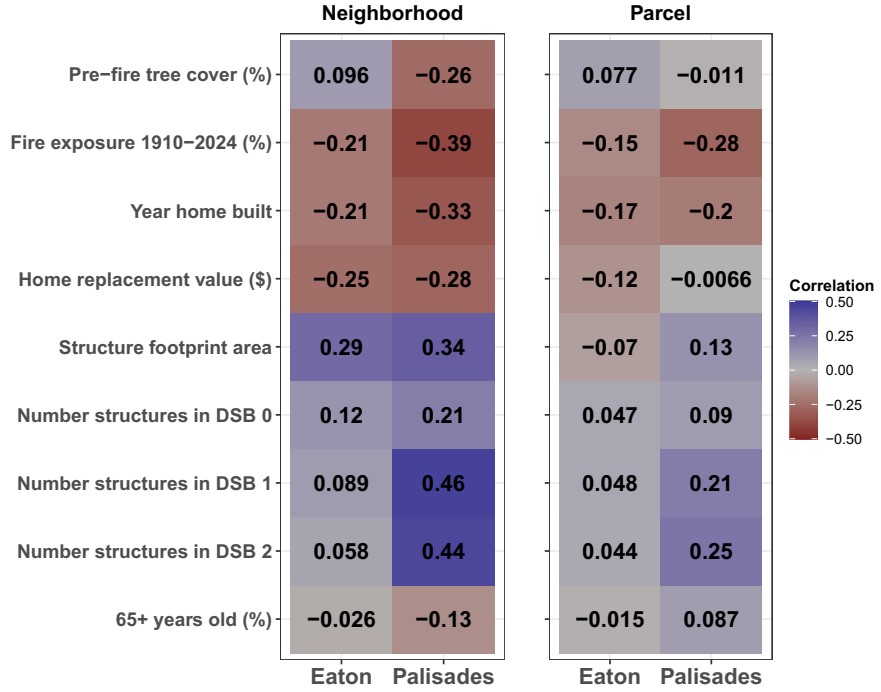

**Fig. 4 | Correlations of socio-ecological characteristics with home destruction at the neighborhood- and parcel-scales.** Correlations are $R^2$ values for the neighborhood scale and pseudo-$R^2$ for the parcel-scale. Red colors represent a negative correlation, blue colors represent a positive correlation, and gray represents a correlation close to zero.

**Table 1 | Neighborhood-Scale Regression Estimates for Predicting Home Destruction (%) in Eaton**

| Standardized Variables | Model 1 (Robust Regression) | | | Model 2 (Generalized Linear Model (GLM)) | | |
|---|---|---|---|---|---|---|
| | Coef. | Robust Std. Err. | $P > t$ | Coef. | OIM. Std. Err. | $P > t$ |
| Home Replacement Value ($) | −0.2234515 | 0.0499102 | 0.000 | −0.2234515 | 0.0572526 | 0.000 |
| Mean Number of Structures in DSB 0* | 0.0790077 | 0.0562508 | 0.161 | 0.0790077 | 0.051119 | 0.122 |
| 1910-2024 Fire exposure (%) | −0.1255753 | 0.0556699 | 0.025 | −0.1255753 | 0.0610845 | 0.040 |
| Structure footprint area (m² ha⁻¹)* | 0.2025663 | 0.0837936 | 0.016 | 0.2025663 | 0.0716475 | 0.005 |
| Median Year Structure Built | −0.0140256 | 0.0668682 | 0.834 | −0.0140256 | 0.0521103 | 0.788 |
| Pre-fire tree cover (%) | 0.2880109 | 0.0599693 | 0.000 | 0.2880109 | 0.0570517 | 0.000 |
| Vacant structures (%) | 0.0819294 | 0.0639504 | 0.201 | 0.0819294 | 0.0700582 | 0.242 |
| Renter occupied structures (%) | 0.0119708 | 0.0425464 | 0.779 | 0.0119708 | 0.0403024 | 0.766 |
| Hispanic (%) | 0.0172493 | 0.0467924 | 0.713 | 0.0172493 | 0.0477052 | 0.718 |
| 65 years and over (%) | 0.022909 | 0.0470383 | 0.627 | 0.022909 | 0.0427722 | 0.592 |
| African American (%) | 0.1811606 | 0.0546205 | 0.001 | 0.1811606 | 0.0451894 | 0.000 |
| Non-English speakers (%) | −0.2275388 | 0.0371709 | 0.000 | −0.2275388 | 0.0360875 | 0.000 |
| Associate degree (%) | 0.1196434 | 0.0398446 | 0.003 | 0.1196434 | 0.0333062 | 0.000 |
| Bachelor's degree (%) | 0.2131117 | 0.0451243 | 0.000 | 0.2131117 | 0.0409068 | 0.000 |
| Below poverty (%) | 0.1149543 | 0.0583681 | 0.050 | 0.1149543 | 0.0479219 | 0.016 |
| Structures built after 2008 (%) | −0.4057521 | 0.0716504 | 0.000 | −0.4057521 | 0.1258849 | 0.001 |
| Per capita income | −0.2566121 | 0.1023824 | 0.013 | −0.2566121 | 0.0952843 | 0.007 |
| Constant | −0.2585042 | 0.0766142 | 0.001 | −0.2585042 | 0.0796636 | 0.001 |
| Log likelihood | | | | −251.3434976 | | |
| AIC | 538.687 | | | 1.754681 | | |
| BIC | 605.7703 | | | −1562.635 | | |
| R squared | 0.529 | | | | | |
| Mean VIF | 1.79 | | | | | |

Model 1 is a robust regression with heteroskedasticity-consistent standard errors. Model 2 is a generalized linear model (GLM). All hypothesis tests are two-sided with α = 0.05, and exact *p*-values are reported without adjustment for multiple comparisons. Data sources and covariate details are described in the Methods and Supplementary Tables 1 and 2. Table Notes: * indicates that all structures from Microsoft building footprints were included, not just homes; *Coef* Coefficient; *Std. Err* Standard Error; *DSB* Defensible Space Buffer; *AIC* Akaike's Information Criteria; *BIC* Bayesian Information Criteria; *VIF* Variance Inflation Factor.

Zones 0, 1, and 2 were positively associated with destruction ($p < 0.001$, $p < 0.003$, $p = 0.076$).

Our Palisades parcel-scale model (Table 4) shows that the median year a home was built was significantly and negatively associated with destruction ($p = 0.002$) and that home replacement value had a strong negative effect ($p < 0.001$). Structure footprint area was also strongly negatively associated with destruction ($p < 0.001$), and single-residence homes were significantly more likely to be destroyed ($p < 0.001$). We found pre-fire UTC was negatively associated with destruction but not statistically significant, while 1910–2024 fire exposure was negatively associated with destruction ($p < 0.001$). Similarly, the number of structures in DSB Zones 0, 1, and 2 were significantly and positively associated with destruction ($p < 0.001$). Total occupants in each home were also positively associated with home destruction ($p < 0.001$).

## Discussion

Our findings show that socio-ecological characteristics such as urban density metrics, socio-demographics, and scale are key in parsing out the complex dynamics of the 2025 L.A. Fires for the Palisades and Eaton communities. At the neighborhood-scale, socio-ecological factors were more highly correlated with home destruction in Eaton ($R^2 = 0.53$) than in Palisades ($R^2 = 0.47$). Conversely, at the parcel-scale, home destruction was better explained by socio-ecological factors in Palisades (pseudo-$R^2 = 0.10$) than in Eaton (pseudo-$R^2 = 0.03$). Previous research has focused on mostly ecological, and more recently socio-demographic, factors related to WUI fire impacts[44,48,49] and such studies are either presented at the neighborhood- or structure-scale. Conversely, our results integrate urban morphology metrics as well as

sociodemographic and fire impact data (i.e., exposure to fire, home destruction) to better understand the multi-scale socio-ecological complexity of these events and their role in: community exposure and vulnerability to urban fires[18,27], WUI mapping and modeling[7], urban fire spread models[8], and improving other fire-related socio-ecological assessments[45,50]. Further, our research also provides a nuanced comparison of the urban fire impacts in two socio-demographically and ecologically disparate communities (Tables 1–4).

At the neighborhood level, the rate of home destruction was high in both Eaton and Palisades (70%, 60% respectively), but the relationships among urban morphology, socio-demographic correlates and fire impact severity varied between the two communities. For Eaton, the proportion of residents who were African American was positively associated with greater destruction, while the proportion of non-English speakers was negatively associated with destruction (Table 1). For Palisades, greater proportions of Hispanic residents, as well as more people with lower educational attainment, were positively associated with destruction (Table 2). For both communities, greater per capita income was negatively associated with destruction (Tables 1, 2). These findings are consistent with recent observed increases in WUI and urban fire-related impacts to communities with greater proportions of non-White residents[1,18,27].

Neighborhood-level urban morphology characteristics related to building density and characteristics were consistent with previous literature[7,8,41], while pre-fire UTC in the two communities had a contrasting relationship with destruction. In Palisades we found more severe fire related destruction in neighborhoods with lower pre-fire UTC, while Eaton showed the opposite relationship (Fig. 3b, Tables 1–4). Interestingly, both Palisades and Eaton had higher UTC

**Table 2 | Neighborhood-Scale Regression Estimates for Predicting Structural Destruction (%) in Palisades**

| Standardized Variables | Model 1 (Robust Regression) | | | Model 2 (Generalized Linear Model (GLM)) | | |
|---|---|---|---|---|---|---|
| | Coef. | Robust Std. Err. | P > t | Coef. | OIM. Std. Err. | P > t |
| Home Replacement Value ($) | 0.0309436 | 0.1132571 | 0.785 | 0.0309436 | 0.0741559 | 0.676 |
| Mean Number of Structures in DSB 0* | 0.2873414 | 0.1267949 | 0.024 | 0.2873414 | 0.1068995 | 0.007 |
| 1910-2024 Fire exposure (%) | −0.1364904 | 0.0883243 | 0.124 | −0.1364904 | 0.0720035 | 0.058 |
| Structure footprint area (m² ha⁻¹)* | −0.0840179 | 0.1104498 | 0.448 | −0.0840179 | 0.0737128 | 0.254 |
| Median Year Structure Built | −0.2746986 | 0.1025601 | 0.008 | −0.2746986 | 0.092456 | 0.003 |
| Pre-fire tree cover (%) | −0.1388522 | 0.1010817 | 0.171 | −0.1388522 | 0.07461 | 0.063 |
| Vacant structures (%) | 0.0457024 | 0.0698876 | 0.514 | 0.0457024 | 0.073237 | 0.533 |
| Renter occupied structures (%) | −0.0704146 | 0.081365 | 0.388 | −0.0704146 | 0.061212 | 0.250 |
| Hispanic (%) | 0.332271 | 0.148115 | 0.026 | 0.332271 | 0.185737 | 0.074 |
| 65 years and over (%) | −0.006193 | 0.0068198 | 0.365 | −0.006193 | 0.0077026 | 0.421 |
| 20–64 years (%) | −0.1607746 | 0.0940146 | 0.089 | −0.1607746 | 0.1033701 | 0.120 |
| White (%) | 0.1379891 | 0.1557884 | 0.377 | 0.1379891 | 0.1682297 | 0.412 |
| Non-English speakers (%) | −0.3934432 | 0.1451512 | 0.007 | −0.3934432 | 0.1276728 | 0.002 |
| Associate degree (%) | 0.4654096 | 0.1409034 | 0.001 | 0.4654096 | 0.1301608 | 0.000 |
| High school (%) | 0.2437277 | 0.1295276 | 0.061 | 0.2437277 | 0.1335921 | 0.068 |
| Structures built after 2008 (%) | 0.1917263 | 0.1193937 | 0.11 | 0.1917263 | 0.1086548 | 0.078 |
| Per capita income | −0.1801959 | 0.0954554 | 0.06 | −0.1801959 | 0.0827865 | 0.030 |
| Constant | 0.8520867 | 0.2962744 | 0.004 | 0.8520867 | 0.2415686 | 0.000 |
| Log likelihood | | | | −235.7809441 | | |
| AIC | 507.5619 | | | 2.169068 | | |
| BIC | 569.7577 | | | −1075.562 | | |
| R squared | 0.4679 | | | | | |
| Mean VIF | 1.97 | | | | | |

Model 1 is a robust regression with heteroskedasticity-consistent standard errors. Model 2 is a generalized linear model (GLM). All hypothesis tests are two-sided with α = 0.05, and exact p-values are reported without adjustment for multiple comparisons. Data sources and covariate details are described in the Methods and Supplementary Tables 1 and 2. Table Notes: * indicates that all structures from Microsoft building footprints were included, not just homes; *Coef* Coefficient; *Std. Err* Standard Error; *DSB* Defensible Space Buffer; *AIC* Akaike's Information Criteria; *BIC* Bayesian Information Criteria; *VIF* Variance Inflation Factor.

**Table 3 | Eaton parcel-scale logistic regression results predicting fire impacts to homes (>50 % damage) using urban morphology, socio-demographic, and home-parcel variables**

| Standardized Variables | Coef. | Std. Err. | z | P > z | [95% conf. Interval] | |
|---|---|---|---|---|---|---|
| Occupants over 65 yrs | −0.0307708 | 0.024581 | −1.25 | 0.211 | −0.078949 | 0.017407 |
| Total occupants | −0.0065007 | 0.043276 | −0.15 | 0.881 | −0.0913199 | 0.078319 |
| Home replacement value ($) | −0.2626829 | 0.052508 | −5 | 0.000 | −0.3655965 | −0.15977 |
| Pre-fire urban tree cover (%) | 0.1057648 | 0.022918 | 4.61 | 0.000 | 0.0608456 | 0.150684 |
| Number of structures in DSB 0* | 0.0825929 | 0.020802 | 3.97 | 0.000 | 0.0418219 | 0.123364 |
| Number of structures in DSB 1* | 0.0674551 | 0.0228 | 2.96 | 0.003 | 0.0227686 | 0.112142 |
| Number of structures in DSB 2* | 0.0409833 | 0.023136 | 1.77 | 0.076 | −0.0043626 | 0.086329 |
| Year home built | −0.2677696 | 0.098066 | −2.73 | 0.006 | −0.459976 | −0.07556 |
| Structure footprint area (m² ha⁻¹)* | −0.2158047 | 0.033813 | −6.38 | 0.000 | −0.2820777 | −0.14953 |
| 1910–2024 Fire exposure (Y/N) | −0.2579235 | 0.024299 | −9.4 | 0.000 | −0.3116851 | −0.204162 |
| Single-family home (Y/N) | −0.1534485 | 0.0231482 | −6.63 | 0.000 | −0.1988183 | −0.1080788 |
| Constant | −0.0064657 | 0.024387 | −0.27 | 0.791 | −0.0542634 | 0.041332 |
| Log Likelihood | −6982.4907 | | | | | |
| AIC | 13988.98 | | | | | |
| BIC | 14076.06 | | | | | |
| Pseudo R² | 0.0306 | | | | | |

Results are from a logistic regression model predicting fire impacts on homes. All hypothesis tests are two-sided with α = 0.05, and exact p-values are reported without adjustment for multiple comparisons. Data sources and covariate details are described in the Methods and Supplementary Tables 1 and 2. Table Notes: * indicates that all structures from Microsoft building footprints were included, not just homes; *Coef* Coefficient; *Std. Err* Standard Error; *z* Z-score; *DSB* Defensible Space Buffer; *AIC* Akaike's Information Criteria; *BIC* Bayesian Information Criteria.

**Table 4 | Palisades parcel-scale logistic regression results predicting fire impacts to homes ( > 50 % damage) using urban morphology, socio-demographic, and home-parcel impact variables**

| Standardized Variables | Coef. | Std. Err. | z | P > z | [95% conf. Interval] | |
|---|---|---|---|---|---|---|
| Occupants over 65 yrs | −0.012541 | 0.026149 | −0.48 | 0.632 | −0.0637927 | 0.038711 |
| Total occupants | 0.3291291 | 0.044931 | 7.33 | 0.000 | 0.2410661 | 0.417192 |
| Home Replacement Value ($) | −0.2733272 | 0.040493 | −6.75 | 0.000 | −0.3526924 | −0.19396 |
| Pre-fire tree cover (%) | −0.0406473 | 0.025638 | −1.59 | 0.113 | −0.0908959 | 0.009601 |
| Year home built | −0.11798 | 0.038155 | −3.09 | 0.002 | −0.1927615 | −0.0432 |
| Number of Structures in DSB 0 | 0.2020334 | 0.02529 | 7.99 | 0.000 | 0.1524666 | 0.25160 |
| Number of Structures in DSB 1 | 0.2072004 | 0.028652 | 7.23 | 0.000 | 0.1510441 | 0.263357 |
| Number of Structures in DSB 2 | 0.3727855 | 0.026875 | 13.87 | 0.000 | 0.3201121 | 0.425459 |
| Structure footprint area (m$^2$ ha$^{-1}$)* | −0.1117505 | 0.024645 | −4.53 | 0.000 | −0.1600532 | −0.06345 |
| 1910–2024 Fire exposure (Y/N) | −0.404121 | 0.0221442 | −18.29 | 0.000 | −0.448314 | −0.3615102 |
| Single-family homes (Y/N) | 0.156179 | 0.0217111 | 7.19 | 0.000 | 0.113626 | 0.198732 |
| Constant | 0.6057343 | 0.026859 | 22.55 | 0.000 | 0.5530916 | 0.6583771 |
| Log Likelihood | −5924.3554 | | | | | |
| AIC | 11872.71 | | | | | |
| BIC | 11958.83 | | | | | |
| Pseudo R$^2$ | 0.1052 | | | | | |

Results are from a logistic regression model predicting fire impacts on homes. All hypothesis tests are two-sided with α = 0.05, and exact p-values are reported without adjustment for multiple comparisons. Data sources and covariate details are described in the Methods and Supplementary Tables 1 and 2. Table Notes: * indicates that all structures from Microsoft building footprints were included, not just homes; *Coef* Coefficient; *Std. Err* Standard Error; *z* Z-score; *DSB* Defensible Space Buffer; *AIC* Akaike's Information Criteria; *BIC* Bayesian Information Criteria.

(23.9% and 29.6% respectively; Supplementary Table 3) compared to other parts of the Los Angeles area (mean of 17.6%)[51]. The differences in the relationship between pre-fire UTC and home destruction in the two communities appear robust when collinearities are considered as they remain when included in multiple regression models (Tables 1–4). However, the differences could be due to differences in tree structure and composition (e.g., tree heights, density, species) or how trees are maintained (e.g., irrigation, pruning) across each community[45]. Similarly, differing firefighter and homeowner actions and neighborhood-scale urban morphology interactions could be factors[45]. Regardless, a more detailed forensic analysis of specific individual parcels and homes would be required to untangle these potential drivers.

Similar to prior studies, we identified that structure age, urban density (structure footprint area), and proximity to other structures (number of structures in DSB Zone 0) were correlated with greater urban fire destruction[8,44,52], and structure value with less destruction[48]. The results for the Palisades fire are consistent with research on urban fires in Australia, which showed that homes with other structures nearby were more likely to be destroyed or damaged[50,53]. Thus, our results show that urban and peri-urban vegetation alone was less important than urban morphology-related factors with regard to home destruction during these two urban fires initiated by wildfires[8,49].

Fire history has historically been associated with changes in urban morphology[24] which could explain the decreased destruction in areas with 1910–2024 fire history. The greater destruction in Palisades for areas with more homes built after 2008 when new building standards were enacted, thus more fire resistant, at first appears to contradict prior research[8]. However, damage seems to be reflected spatially by urban development patterns in Palisades, where newer structures tend to be built in the most affected, high-density southeastern areas of the community (Supplementary Fig. 4) instead of the community edge adjacent to wildland vegetation. These neighborhood-level findings of high home destruction in dense urban communities contrast with the WUI fire literature that reports that lower housing density (i.e., urban periphery, WUI) is associated with greater fire risk[5,54]. Specifically, structure location relative to natural vegetation patches is reported to be an important factor in home destruction during wildfire planning and management[52].

Many correlates of home destruction were consistent between the parcel- and neighborhood-scale analyses, but structure footprint area was a key exception (Fig. 4; Tables 1–4). For each of the communities, the relationship of several other correlates with home destruction varied across the neighborhood and parcel scale, such as home replacement value and number of nearby structures (Eaton), and the year the home was built (Palisades). Indeed, other studies of fire-building loss relationships have found similar differences when presenting these relationships at either the parcel- or neighborhood-scale[40,43,45]. However, our results show that for correlates like structure footprint area, the scale of analyses will determine what factors contribute most to home destruction in urban wildfire-driven events[49,52]. This suggests that parcel- and neighborhood-scale analyses are both important for understanding the structural and human dimensions of urban fire impacts (Fig. 4; Supplementary Table 5).

It is important to consider that there were some limitations in studying the two communities impacted by the 2025 L.A. fires. First, we used only three parcel-scale socio-demographic variables in our analysis and excluded predicted race from modeling due to high uncertainty. So, we encourage readers to use our framework for integrating urban morphology, socioeconomic, and human community information across home and neighborhood scales for future studies, including a wider array of parcel-scale socio-demographic variables. Second, the different-sized analysis units for socio-demographic variables (i.e., U.S. census blocks and block groups) will have potential biases and spatial discontinuities arising from sampling variability and additional inaccuracies resulting from downscaling. That said, data were spatially disaggregated, and generalized linear and robust regression methods weighted by population were applied to mitigate heteroscedasticity and accommodate nonlinear patterns (refer to Methods in Supplementary Information for specifics). We also note that the R$^2$ and pseudo-R$^2$ values for our model fit results were in-line with ranges reported in other research in the wildfire literature[13,41,48]. Finally, our quantitative analyses of the impacts of urban fires did not capture two important groups of factors. Specifically, we did not quantify the role of environmental factors (e.g., wind speed, topography, fire behavior)[22] or proximity of homes to wildland fuel types since these are well studied[4,7] and likely would not have changed our results given the descriptive nature of our analysis. Nor do we model the role of

stochastic actions of firefighters, emergency management professionals, and community members in saving homes during these events. However, we acknowledge that environmental factors, parcel-scale fire suppression actions, urban morphology, and socio-demographic factors could all be important factors in developing predictive models of urban fire impacts.

In fire-prone wildland ecosystems, fire can have both positive and negative impacts on ecosystem services and communities[55]. For example, 'prescribed' fires and fuel treatments near these urbanized communities can play an important role in improving ecosystem health and reducing fire risk in forested ecosystems driven by excess fuels[15,56]. However, these are not often effective in preventing wind-driven urban fires like the Eaton and Palisades fires[13,15]. Such wind-driven urban fires are spread by mostly non-vegetative fuels (i.e. homes, embers, radiant heat) and are typically caused by human or infrastructure-related (i.e., powerlines) ignitions[16–18,22,57]. As a result, urban fires have few positive outcomes as they destroy homes, result in human fatalities, and lead to indirect human health impacts and deaths[58–60].

Therefore, future research on urban fires initiated by wildfires can build on our approach and methods to better understand the complex and interacting dynamics of fire on highly populated urban communities and their vulnerabilities. Given the increasing frequency of these events, integrating multi-scale, socio-ecological approaches to studying the impacts of urban fires on communities is critically needed. Indeed, some socio-ecological, parcel-scale factors used in our analyses (e.g., structure footprint area, number of homes in DSB Zones, urban tree cover) could also be used for developing or improving urban fire behavior models[9,50]. Developing fine-scale downscaling methods that integrate disparate data sources to improve accuracy and spatial resolution, particularly for socio-demographic factors at the parcel-scale, would be an important first step. This is particularly relevant in urban ecosystems since the parcel scale will determine the vulnerability of individuals and families, land tenure, human actions, maintenance practices, building construction, and landscape maintenance regimes of individual homeowners. Accordingly, future research that combines transdisciplinary quantitative analysis with qualitative data elicitation methods is warranted to better capture the social, economic and governance dimensions in these communities affected by urban fires.

Urban fire disasters are occurring more frequently, not only in western North America[1], but globally[61] across various biomes and contexts[2]. As such, there is a need to better understand them in terms of social, environmental and economic costs given their catastrophic nature. We spatially analyze and quantify the socio-ecological characteristics across different scales of two diverse communities, and their inherent neighborhoods and homes, impacted by urban fires. We found that socio-ecological characteristics and scale are key in parsing out the dynamics of urban fire events. Also, urban vegetation is less important for home destruction than other urban morphology factors (i.e., structure footprint area). We also provide a nuanced approach for defining and characterizing fire-impacted communities and urban density metrics at three different scales. This has implications for defining and mapping 'vulnerable communities at risk of fire'. Our findings and analyses of the impacts of the 2025 L.A. urban fires span multiple scales, however due to the limitations of U.S. census data, socio-demographic impacts (e.g., age, race, ethnicity, income, etc.) at the scale of individual parcels are little studied and need to be understood further. While urban fires initiated by wildfires have many parallels with urban conflagrations from the 19th and early 20th centuries, this re-emerging problem requires paradigm shifts in how we study, manage, respond to, and recover from urban fires.

In conclusion, based on our findings that highlight the importance of scale of analysis and account for both biophysical and socio-economic drivers of urban fire impacts, adopting the resilience and adaptive cycles from the socio-ecological systems literature to understand these fire regimes in urban ecosystems and communities could shed light on pre- to post- fire actions. Specifically, thinking of the urban fire cycle as an adaptive process involving pre-fire planning and preparation, firefighting operations and evacuation, and then post-fire response, recovery, and restoration/rebuilding are key and merit future research. For urban fires that behave similarly to the 2025 L.A. fires, considerations across various spatial scales provide a valuable next step towards understanding not only what happens during urban fires and who is impacted and vulnerable, but also how and what policies and management actions can be formulated and successfully implemented to mitigate—and adapt to—the risk of future disasters.

## Methods

### Direct fire impacts
We analyzed the degree of direct fire impacts to individual homes at the neighborhood and parcel level by using California Department of Forestry and Fire Protection (CALFIRE)'s Damage Inspection (DINS) data[62]. We quantified the proportion of homes destroyed at the neighborhood-scale and the number of homes destroyed at the parcel-scale. Structures at the parcel level were mostly single-family homes as well as other minor structures and multifamily residential structures (Supplementary Fig. 1). When considering fire impacts, we only considered impacts to major structures (e.g., homes) as observed by the CALFIRE DINS data collectors and excluded parcels that had only miscellaneous structures (e.g., garages) from the data set. The remaining structures in this data set are referred to as destroyed homes, with the exception of the 'numbers of structures in DSB Zones 0, 1, and 2' and 'structure footprint area variables', which accounted for all fire-impacted structures (Supplementary Tables 1, 2). We use exposure to refer to homes and structures that were inside the fire perimeters, but that were not specifically damaged or destroyed.

### Statistical analysis
To investigate the relationship between fire-related damage and socio-ecological variables at the neighborhood-scale, we used Generalized Linear Models (GLMs) and Robust Regression methods[42,43,63], while at the parcel-scale we used Binary Logistic Regression to model a binary outcome (i.e., home destruction) as a function of continuous predictors[64]. The Robust Regression method[42,43] was used to confirm the results of the GLMs due to non-Gaussian error distributions and the potential influence of outliers at the neighborhood-scale. The Binary Logistic regression was used given its suitability for non-normal error distributions and categorical responses[64]. All statistical analyses were done using STATA software Version 19[65]. When interpreting our results, we used the following definitions for levels of statistical significance: 'Marginally significant' refers to results with $p$-values between 0.05 and 0.10, 'significant' refers to results with $p$-values less than 0.05, and 'strongly significant' refers to results with $p$-values less than 0.01.

### Study design, datasets, and socio-demographic variables
To understand the socio-ecological dynamics and impacts of the 2025 L.A. fires, we used three spatial scales for our analysis: community, neighborhood, and parcel scale. Community refers to the Eaton and Palisades areas affected by the 2025 LA Fires. Neighborhood refers to groups of homes located in U.S. census blocks[66], while parcel refers to single structures or homes in individual properties. For each community we characterized socio-demographic characteristics, urban morphology, and direct fire impacts to homes at both the urban neighborhood- (Supplementary Table 1) and parcel-scales (Supplementary Table 2). We used urban defined U.S. census blocks[66] to represent neighborhoods and L.A. County Assessor's Office parcels to represent individual urban residential parcels. Multi-scale socio-

demographic data was compiled from 2020 U.S. census data, U.S. Army Corps of Engineers structure data, and L.A. County parcel records with the rethnicity model[67] applied to predict race from property owner names (Supplementary Table 1). We used U.S. census block group level socioeconomic data (e.g., income, poverty, education, language proficiency) from the 2019-2023 American Community Survey (ACS) five-year estimates, despite previously reported issues (e.g., spatiotemporal resolutions, marginal and standard errors), because of their use in similar studies[1,28]. For additional details and limitations see the Methods in the Supplementary Information.

### Urban morphology and history

We characterized urban morphology or the biophysical and land use characteristics of the two communities at the neighborhood- (Supplementary Table 1) and parcel-scales (Supplementary Table 2). For each urban neighborhood and parcel, we quantified: 2025 proportion fire exposed, 1910–2024 proportion fire exposed, pre-fire UTC, number of structures in DSB Zones 0, 1 and 2, structure footprint area, and structure replacement value (see Methods in Supplementary Information for details). For both the 2025 fires and 1910-2024 fires, we delineated fire-exposed urban areas using spatial intersection analysis to identify areas in the communities within both the fire perimeters and affected U.S. census blocks[66] or parcels. Pre-fire UTC was calculated using the percentage of UTC in each U.S. census block or parcel. We quantified the number of nearby structures within DSB Zones 0 (0–1.5 m), 1 (1.5–9.1 m), and 2 (9.1–30.5 m), from the main home on each parcel[7], and averaged across all parcels in each neighborhood using the Microsoft building footprint[68] data set. We calculated urban density as structure footprint area (m$^2$ ha$^{-1}$) or the sum of all structure and home footprint area per unit land area, for each U.S. census block[66] and parcel (Table 1).

### Reporting summary

Further information on research design is available in the Nature Portfolio Reporting Summary linked to this article.

## Data availability

All code for data cleaning, analysis, and figure generation, as well as processed data for this study, are available in the following figshare repository: https://doi.org/10.6084/m9.figshare.29936876.

## Code availability

All code for data cleaning, analysis, and figure generation is also available in the following GitHub repository: https://github.com/carlnorlen/la-urban-fires.

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

## Acknowledgments

CA Norlen was funded by the USGS National Land Imaging Program, and FJ Escobedo and S Sharma were funded by the USDA Pacific Southwest Research Station. This manuscript was greatly improved by comments from AR Carlson and OO Drakes. We also acknowledge the indigenous peoples and the land of California. The Los Angeles metropolitan area, where this research took place, is located on the ancestral territories of the Acjachemen, Cahuilla and Tongva peoples, many of whom maintain active physical, spiritual, and cultural ties to the region.

## Author contributions

C.A.N., F.J.E., and S.S. designed the study, reviewed and discussed the results, and revised the manuscript. C.A.N. and S.S. assembled data, wrote code, analyzed data, and created figures and tables. C.A.N. and F.J.E. wrote the initial manuscript draft.

## Competing interests

The authors declare no competing interests.
