## [Transparent Peer Review file · Nature Communications]

Socio-Ecological Impacts of the 2025 Los Angeles Urban Fires on Communities, Neighborhoods, and Homes

Corresponding Author: Dr Carl Norlen

Version 0:

Reviewer comments:

Reviewer #1

(Remarks to the Author)

Authors mentioned in Table S1 that they used the ACS 5-year estimates from 2020 or 2023, but it is not entirely clear whether these estimates were applied consistently across all datasets used from ACS. Similarly, for example, if the Microsoft building footprint dataset from 2020 (Table S1) was used for calculating the structure of the basal area, there is an important temporal limitation that should be acknowledged. The Microsoft building footprint data represents a static snapshot of 2020 and does not reflect subsequent losses or changes due to wildfire events after 2020. Consequently, structures destroyed in fires occurring in 2021–2024 would still appear in the dataset, even though they no longer exist in the real world and the rebuilt information of houses is also missing. This temporal mismatch could lead to an overestimation of building density and structure counts in recently burned areas when calculating metrics such as “structures per unit land area” and other indicators. From a hazard assessment perspective, this introduces potential bias in spatial analyses, particularly if the intent is to characterize current or post-event conditions rather than pre-fire baselines. In addition, the footprint data counts all structures, including, commercial, residential, and other small structures attached or nearby main house; authors did not include details of this limitation and how they cleaned the data for their purpose in this study.

Furthermore, if ACS estimates are used in combination with static building footprint data, the differing temporal resolutions and update frequencies of these datasets could compound these uncertainties. The authors should discuss the implications of these mismatches and, if possible, quantify their potential effect on results. There is a lack of clear quantification on how factors such as urban and peri-urban vegetation are compared to urban morphology and what their relative importance is. The 2025 LA fires were highly driven by severe weather conditions, such as heavy wind gusts and thus these factors should be also explored in addition to socio-ecological factors. The location of these structures also plays an important role—its proximity to flammable vegetation, topography—in determining where wildfire will survive or not at all and that is needed to be analyzed to capture the overall picture, specifically for recent LA fires. The term “urban fire cycle” as introduced in the conclusion (lines 382–386) as well in the abstract section appears vague and currently lacks sufficient definition in existing fire science literature. While the authors’ intention to conceptualize a set of linked processes, i.e., from pre-fire planning through to post-fire recovery is clear, the use of the word “cycle” implies a systematic, perhaps ecological recurrence that may not be appropriate accurate in 2025 urban fire in LA and overstate the coherence of what appears to be a procedural management sequence.

(Remarks on code availability)

Reviewer #2

(Remarks to the Author)

The following manuscript was reviewed: “Socio-Ecological Impacts of the 2025 Los Angeles Urban Fires on Communities, Neighborhoods, and Homes”. The authors explain that their objectives are to look into which socio-ecological factors were associated across the two communities impacted by the 2025 LA County Fires and then determine how the relationships varied based on the scale considered (neighborhood vs. parcel level). The topic covered is of growing importance and works to understand dynamics of who may be most impacted by these wildfire turned urban fire events though the general scope of reference of the results of this work is limited to the two fires analyzed the methods could be used to work to understand other fires and their impacts.

Line 110 you state that three spatial scales were used for analysis: community, parcel, and neighborhood. I think it would be helpful to give more information on how each of these scales were defined in text even though more detail is given in the supplemental information.

Line 118: for the neighborhood scale analysis you state you used GLMs and robust regression techniques. Can you explain what these robust regression techniques were for those who may not be familiar? Also, please provide reasoning for why these analysis techniques were chosen. What benefit do they provide?

Line 119: You switched analysis models for the parcel scale to binary logistic regression. Provide logic as to why you did this? Why did the different scales require different modeling techniques?

Line 130-132 I know more detail is provided in the supplemental information, but I think it would be helpful to state how you did this in the main body of the text so the reader does not constantly have to flip between documents.

Line 132: Is structural basal area a term you created to link the urban environment to the wildland since basal area refers to the cross-sectional area of trees at breast height? To me, this just feels like structural density.

Line 175-177 (Figure 2 Caption) I would put the (a) and (b) before the fire you are referencing (e.g. (a) Palisades and (b) Eaton) as that is the typical convention. It took me two read throughs to figure out what your (a) and (b) were referencing at first.

Figure 3. Your R² values are all quite small but most of your p values indicate there is statistical significance to your results. I am just curious as to if you tried any other model types to see if the R² values improved.

Line 206: this is a line of the main manuscript that separates Figure 4 and its caption

In discussing the impacts of different factors you frequently used the terms significantly but marginally, marginally significant, significant, and strongly but I did not see where these terms were defined. What is the difference between each/what are their cutoffs?

Line 252-254 Can you further explain the property owner race results you were comparing to the census data? Were these self-reported?

Line 324 "the most affected, higher density southeastern areas of the community" I was confused by what this referred to. Are you saying that newer homes tended to be in areas with greater impacts from these fires (e.g. the edges of communities?)

Line 325-326 Not all WUI fire literature report that lower housing densities are associated with greater fire risk. For example, I know off the top of my head that Caggiano et al (2020) reported that intermix WUI areas experienced the highest rates of housing loss to wildfire versus interface and that building loss mostly occurs in areas with low building density and high vegetation cover.

I like that the authors clearly stated the limitations of their analysis (lines 340-353)

In general, I think the work presented in this paper is needed and a step forward in understanding how fires impact communities; however, I personally would like more detail to be provided in the methodology. With the current level of detail provided, I am unsure I would be able to properly replicate the processes were I to try.

I think it would be helpful to incorporate more of the supplemental information into the main body of text. Additionally, I think the results could focus more on the story they are telling. I think more of the why are the findings important should be included when discussing the analysis results.

(Remarks on code availability)

I did not see that any code was provided.

Reviewer #3

(Remarks to the Author)

The manuscript "Socio-Ecological Impacts of the 2025 Los Angeles Urban Fires on Communities, Neighborhoods, and Homes" provides a detailed spatial and statistical analysis of the destruction caused by wildfire-initiated urban fires in two contrasting communities in Los Angeles.

Comments:

- Can the authors please clarify the concept of "wildfire-initiated urban fires"? I assume that the authors mean wildfires that affect, reach or enter urban areas/cities, but this is not clear. I understand your point-of-view, expressed at the end of page 3 and beginning of page 4, but I believe that "wildfire-initiated urban fires" is a confusing terminology. On the other hand, "urban fires" are mentioned in the title and "wildfire" is one of the keywords. An urban fire is usually associated with a human trigger/cause/fault (e.g. short circuit), having no connection to natural causes, especially to wildfires. So, there are different

mechanisms and rational intervening here. At the same time, the authors mention “wildfires”, but writes the following: “As such, events like the 2025 Eaton and Palisades fires in L.A. have been deemed wildfire initiated urban fires; not wildfires”. In sum, the authors need to clarify these terms and their correct use.

- Following the previous comment, what are the differences in fire behavior in this event (caused by a wildfire), compared to an event caused by a short circuit?
- Page 4, lines 70-72: “Key to this is who lived in these communities and what types of homes, neighborhoods, and environmental amenities were lost or damaged as these factors”. This configures an exposure and vulnerability study. However, these terms are rarely used in this manuscript, while these should be a focal point of the study.
- The abstract lacks information regarding the data and methods used in the manuscript. This should be added to the abstract. Additionally, the results mentioned in the abstract are a little vague. It is necessary to improve these points.
- The study area section must include a map representing the location of the studied communities/neighborhoods. Information regarding the size/area of Eaton and Palisades should be included in this section. The authors should also include more information regarding: 1) the meteorological conditions before and during the event; 2) the rainfall amounts during 2024 (spring, summer and autumn months). Lastly, what was the proportion of buildings affected by fires during this event?
- In data and methods section, the authors mentioned “robust regression techniques”. This should be properly explained.
- Have the highly correlated independent variables been eliminated?
- At the beginning of page 7, this is mentioned: “the proportion of homes destroyed by fire in each neighborhood and the number of homes destroyed in each parcel”, but it is not clear which is/are the dependent variables used.
- The methods used in this study should be better explained. I suggest that the authors build a flowchart with the steps followed.
- Figure S1 should be included in the text. This is important to understand the differences between the two study areas.
- Figure 1 leaves much to be desired. The lines are too thick, and it is unclear what values are reached in 2025.
- Page 7, lines 159-160: “About 30% of Palisades was impacted by historic fires from 1910-2024, while fires in 1938 and 1993 alone impacted >25% of the community (Figure 1)”. This sentence is confusing. The authors mentioned 30% between 1910 and 2024. This value came from where? It is impossible that this the average value in this period. On the other hand, this chart somehow contradicts past sentences (e.g. “Such urban conflagrations were historically common in cities prior to the 1900s or during periods of armed conflict”). The high peak in the chart show that these events were not uncommon and new in Palisades (despite the extreme event of 2025).
- Figure 2 needs to be improved. The lines of the polygons are too thick (red and blue). There many areas covered in black. This means that there were no houses destroyed, despite these were affected by fires? There are some isolated areas/polygons with no connection with the other polygons. The readers will not understand what happens between these polygons/areas. Forests? What is the relationship between the proportion of houses destroyed and the distance to the polygons’ edges? This should be quantified.
- Figure 3 only presents some independent variables and mixes urban morphology and sociodemographic characteristics. Why did the authors choose these 6 variables? An explanation is needed in the text.
- Page 9, line 181: “exposure” should be replaced by affected. If I understand correctly, these houses were affected/hit by previous fire events. So, it is more correct the use of affected.
- Figure 4 should present the correlation coefficients of all variables considered in both study areas.

(Remarks on code availability)

Version 1:

Reviewer comments:

Reviewer #2

(Remarks to the Author)

The following manuscript was reviewed for the second round of peer review: “Socio-Ecological Impacts of the 2025 Los Angeles Urban Fires on Communities, Neighborhoods, and Homes”. In general, I think the manuscript has been improved; however, in giving the paper a read through I still think there are some general clarity and proof-reading issues. I believe the topic covered is of growing importance and works to understand dynamics of who may be most impacted by these wildfire turned urban fire events though the general scope of reference of the results of this work is limited to the two fires analyzed the methods could be used to work to understand other fires and their impacts. The changes to the manuscript did make the methods more clear and replicable and less reliant on the supplemental information. Below my direct comments are outlined:

- Line 33: Money formatting is non-traditional. I would suggest using \$76-131 billion USD.
- Line 37: colon is not needed
- Line 39-45: Why are you putting the terms communities and structures in ""? This is not done elsewhere in the paper.
- Line 61-66: This sentence is very long and confusing. I had to read it multiple times before I understood what you were saying. Consider splitting it up into more than one sentence for clarity.
- Line 81 : em-dash around “and scales” is not needed
- Line 103-105: Once again non-traditional money formatting and estimated used twice. Consider rewording to reduce repetition.
- Line 106-108: Common separated list of areas was confusing. Consider doing: “...including Altadena, Pasadena, and Sierra Madre area (Eaton hereafter); Pacific Palisades (Palisades hereafter); and adjacent wildland areas in the San Gabriel and Santa Monica Mountains.” or something similar.

- Line 158-160: Glad you included more information on what structural basal area is BUT I would like to see WHY this metric is better than structure density which is what is commonly accepted and used in literature. What benefit does this new metric provide?
- Line 171-172: How did you determine which structures were "major" and which were miscellaneous? By area?
- Line 195: 100+ years? This is confusing. Should it just be 100 years?
- Line 201-202: Why were these chosen for analysis? I want more information that they have also be reported in other literature.
- In the discussion I would like to see more of the story being told. Why are these finding significant and what do we gain from them? This was hit on some but I feel like more could be done. I felt that the claims regarding the importance of urban morphology over vegetation, scale-dependent relationships, and sociodemographic disparities in impacts were well supported, but I'd love to see more information and discussion regarding the mechanism behind opposing UTC relationships between communities and how the omission of environmental factors (wind, topography, fire behavior, etc.) could impact results.
- Line 420: Include a citation for "Urban fire disasters are occurring more frequently and globally"
- Line 427-429: This feels a little vague for conclusions. What specifically could use further study and why? How can it build upon your work?
- Line 433-438: How does this connect to your findings?
- In conclusions would be good to synthesize key finds. The conclusions covers novelty of approach before jumping to future frameworks without reminding us of what your knowledge your study created.
- Line 431: Typo. "paradigm shifts" should be "paradigm shifts"

(Remarks on code availability)

Reviewer #3

(Remarks to the Author)

After the changes made by the authors, the manuscript can be accepted in its present form.

(Remarks on code availability)

Version 2:

Reviewer comments:

Reviewer #2

(Remarks to the Author)

Thank you to the authors for taking the time to thoughtfully address my previous comments. I feel that the questions I posed were sufficiently answered and the text changes greatly helped increased the clarity of the manuscript. I believe the current manuscript can be accepted in this revised form.

(Remarks on code availability)

RESPONSE TO REVIEWER COMMENTS

Dear Reviewers,

Thank you for your careful review of our manuscript. As requested, we have done a major revision of the manuscript, and we believe the updated version is greatly improved. We made major revision to the methods, improved the use of terminology throughout the manuscript, and revised figure and tables to improve clarity. In response to comments from all three Reviewers we revised the methods section by moving text from the Supplementary Materials to the main text and improving the clarity of language throughout. As requested by Reviewer #3 we made major revisions to Figure 1 including an additional panel (a) to highlight the study region and make the research approach clearer.

In addition, in response to a comment from Reviewer #1 we clarified our use of “fire exposure” throughout and “urban fire cycle” in the discussion. We also revised the abstract to better highlight the data and methods and results as requested by Reviewer #3. We have also revised the Results to better highlight the story as requested by Reviewer #1. In addition, we revised and addressed comments about Figures 2, 3, and 4 to improve their clarity for readers as suggested by Reviewers #2 and #3. Finally, we added additional references for context and additional background throughout the manuscript and revised the text throughout to improve clarity.

Please find detailed responses to these and additional comments below with our responses in italics.

Sincerely,
Carl Norlen, PhD (corresponding author)

Reviewer #1 (Remarks to the Author):

Authors mentioned in Table S1 that they used the ACS 5-year estimates from 2020 or 2023, but it is not entirely clear whether these estimates were applied consistently across all datasets used from ACS.

We appreciate the reviewer’s attention to our methods and in this revision, we have now clarified our approach. Specifically, all block-level demographic variables were obtained from the 2020 Decennial Census, which offers detailed population data at the smallest geographic scale. For variables not available at the Block level—such as socioeconomic factors—the 2019–2023 ACS 5-year Block-Group estimates (released in 2023) were used. Only these two data sources were used in the analysis; no other ACS datasets or release years were included for consistency. We have updated the methods to clarify this point. The revised text at lines 139 to 142 is as follows, “We used U.S. census block group level socioeconomic data (e.g., income, poverty, education, language proficiency) from the 2019-2023 American Community Survey (ACS) five-year estimates, despite previously reported issues (e.g., spatiotemporal resolutions, marginal and standard errors) because of their use in similar studies ^{1,28}.” We hope this addresses your concern.

Similarly, for example, if the Microsoft building footprint dataset from 2020 (Table S1) was used for calculating the structure of the basal area, there is an important temporal limitation that should be acknowledged. The Microsoft building footprint data represents a static snapshot of 2020 and does not reflect subsequent losses or changes due to wildfire events after 2020. Consequently, structures destroyed in fires occurring in 2021–2024 would still appear in the dataset, even though they no longer exist in the real world and the rebuilt information of houses is also missing. This temporal mismatch could lead to an overestimation of building density and structure counts in recently burned areas when calculating metrics such as “structures per unit land area” and other indicators. From a hazard assessment perspective, this introduces potential bias in spatial analyses, particularly if the intent is to characterize current or post-event conditions rather than pre-fire baselines. In addition, the footprint data counts all structures, including, commercial, residential, and other small structures attached or nearby main house; authors did not include details of this limitation and how they cleaned the data for their purpose in this study.

As pointed out by the reviewer, indeed, the building footprint data from Microsoft represents a static representation of the pre-fire conditions of buildings before the fire in 2020. According to our analysis of the fire history, no fires occurred in our study areas

between 2021-2024 (see Figure 1), so there should be no decrease in building footprint density due to fire impacts. But, it is also possible that there were other changes to building density from urbanization and construction between 2021-2024 which would lead to an underestimation of building density in the study areas. However, we assumed these changes were negligible. While the building footprint data set closely aligns with the 2020 Decennial Census, there are limitations such as changes in the arrangement of structures between 2020 and 2024 similar to the potential changes to socio-demographic variables between the date of the US Census data and the date of the fire. We have also added text to the Supplementary Materials to address the limitations of the building footprint data. The edited text at lines 153-158 is as follows, “We calculated urban density as structure basal area ($m^2 ha^{-1}$) or the area of fire-affected homes and structures, or the sum of all structure and home structure-home footprint area, for each U.S. Census Block and parcel in the fire affected communities (Table 1). Structure basal area is a novel metric that measures not only the density of structures per unit area but also their footprint area similar to forest basal area to provide a more nuanced metric of the role of urban density in urban fires.” We hope this addresses your concerns.

Furthermore, if ACS estimates are used in combination with static building footprint data, the differing temporal resolutions and update frequencies of these datasets could compound these uncertainties. The authors should discuss the implications of these mismatches and, if possible, quantify their potential effect on results. There is a lack of clear quantification on how factors such as urban and peri-urban vegetation are compared to urban morphology and what their relative importance is.

We will address your comments in 3 parts, first the building footprint data issues, then uncertainties around the use of ACS data, and then on the importance of urban versus peri-urban vegetation on impacts.

As explained in the previous comment, the building footprint data from Microsoft represents a static representation of the pre-fire conditions of buildings before the fire in approximately 2019-2020 according to the dataset creators: GitHub - microsoft/USBuildingFootprints: Computer generated building footprints for the United States. According to our analysis of the fire history, no fires occurred in our study areas between 2020-2024, so there should be no change in building footprints due to fire, but changes could occur due to other events such as urbanization, remodeling, additional dwellings constructed in the parcel, etc. While the building footprint data set closely aligns with the 2020 Decennial Census, there are limitations such as changes in the arrangement of structures between 2020 and 2024 similar to the potential changes to socio-demographic variables between the date of the US Census data and the date of the fire. We

have also added text to the Supplementary Materials to address the limitations of the building footprint data. The edited text at lines 49 to 55 in Supplementary Materials is as follows, “The Microsoft building footprint data represents a static snapshot of building footprints for 2019-2020, we assumed this is an accurate pre-fire representation of building footprint for Eaton and Palisades which ignores any buildings constructed or removed from 2021-2024. We believe that this is a reasonable assumption as no fires occurred within either community from 2021-2024 (Figure 1b).” We hope this addresses your concerns.

Regarding the ACS data, you are correct in that there are several limitations and these include: issues with the spatial and temporal resolutions, uncertainties in smaller-rural census tracts, small sample sizes, modifiable areal unit problems, and higher marginal and standard errors. Although we do account for issues of spatial uncertainty as outlined in the Spatial disaggregation section of our Supplementary methods, there are other limitations that cannot be normalized. For a list of these, please see refer to this document: https://www2.census.gov/programs-surveys/acs/tech_docs/data_suppression/ACS_Data_Release_Rules.pdf

However, this is the best available data that can be used to address our research questions and several other relevant studies, some of which we cite, have used the data including:

- Yadav, Kamini, et al. "Increasing wildfires and changing sociodemographics in communities across California, USA." *International Journal of Disaster Risk Reduction* 98 (2023): 104065.
- Davies, Ian P., et al. "The unequal vulnerability of communities of color to wildfire." *PloS one* 13.11 (2018): e0205825.
- Masri, Shahir, et al. "Disproportionate impacts of wildfires among elderly and low-income communities in California from 2000–2020." *International journal of environmental research and public health* 18.8 (2021): 3921.

So, to address your concerns we now specify in the main text the use of the ACS data and its issues in the limitation section of our Discussion. Specifically we now state, “Second, the different sized analysis units (i.e., US Census Blocks and Block Groups) will have potential biases and spatial discontinuities arising from sampling variability, and additional inaccuracies (e.g., modifiable area unit problem, ecological fallacy) resulting from downscaling and spatial overlaps. Additionally, we refer the reader to our Supplementary methods section for specific details and the limitations related to the ACS data. First, we have revised the Study Design, Datasets, and Socio-Demographic Variables section by adding this sentence and modifying the text and it now states, “ We used Block group level socioeconomic data (e.g., income, poverty, education, language proficiency)

from the 2019-2023 American Community Survey (ACS) five-year estimates, despite previously reported issues (e.g., spatiotemporal resolutions, marginal and standard errors) because of their use in similar studies ^{1,28}. For specific details and limitations see the Supplementary methods section.” Second, in the Sociodemographic data section of our Supplementary methods we add this sentence at the end, “We do note the limitations in the ACS data (i.e., tradeoffs in spatial-temporal resolutions, higher marginal and standard errors), however this is the best available data and has been used in other similar studies such as Davies et al., (2018) and Yadav et al., (2023).” We hope this addresses your concerns.

Finally regarding “.. a lack of clear quantification on how factors such as urban and peri-urban vegetation are compared to urban morphology and what their relative importance is”; we do note an important oversight on our part that might have led to your confusion. Specifically, we did not specify that we only analyzed US Census Bureau defined “urban” Blocks. Therefore peri-urban (rural), non-urban, block groups were not included in our analyses. To clarify this we have inserted “urban” in the following 4 sentences in the Study Design, Datasets, and Socio-Demographic Variables and Urban Morphology sections (see underlined term):

- “...direct fire impacts to homes at both the urban neighborhood (Table S1) and parcel scales (Table S2).”
- “We used urban defined U.S. Census Blocks ³⁷ to represent neighborhoods and L.A. County Assessor’s office parcels to represent individual urban residential parcels.”
- “For each urban neighborhood and parcel we quantified..”
- “..we delineated fire-affected urban areas using spatial intersection analysis to identify areas in the communities..”

There were however areas and patches of wildland/peri-urban/WUI type vegetation in our urban analysis units. These same types of patches of “light flashy fuels” are indeed where fire ignition occurred on both fires. However, our on-the-ground experience and conversations with colleagues and firefighters who responded to and were on the incident, and are actively working on fire management in the affected areas, mentioned that wildland vegetation had little to do with fire behavior and damage in our urban study area. In fact, one Incident Commander from the Eaton fire noted that fire spread from the ignition source in nearby wildland vegetation fuels to urban structures and multi-family homes within 10 minutes.

Since we cannot include such personal communications, we instead summarize this by revising the second to last paragraph in our Discussion section with new text. The

revised paragraph now reads, “Fire in fire-prone wildland ecosystems can have both positive and negative impacts on ecosystem services and communities⁴⁹. For example, “prescribed” fires and fuel treatments near these urbanized communities can play an important role in improving ecosystem health and reducing fire risk in forested ecosystems driven by excess fuels^{14,48}. However, these are not often effective in preventing wind driven urban fires like the Eaton and Palisades fire^{12,14}. Such wind-driven urban fires are spread by mostly non-vegetative fuels (i.e. homes, embers, radiant heat) and are typically caused by human or infrastructure-related (i.e., powerlines) ignitions^{15-17,49}. Conversely, urban fires have little positive outcomes as they destroy homes, result in human fatalities, and lead to indirect human health impacts and deaths⁵⁰⁻⁵².

The 2025 LA fires were highly driven by severe weather conditions, such as heavy wind gusts and thus these factors should be also explored in addition to socio-ecological factors. The location of these structures also plays an important role—its proximity to flammable vegetation, topography—in determining where wildfire will survive or not at all and that is needed to be analyzed to capture the overall picture, specifically for recent LA fires.

You are correct in that these recent and more destructive urban (and wild) fires we present in our introduction and study aim, do co-occur with high winds and increased human related ignitions. Studies we cite (Purnomo et al. 2024; Syphard et al., 2012), and many others including Balch et al., (2024) document this. However, we consider wind an ‘environmental’ factor and hence outside the scope of study. However, to address this concern, we now specify this as a limitation of our study; we will shortly specify how we do so.

Similarly, topography is another ‘environmental’ factor we do not study. We do not do so since Purnomo et al. (2024), which we cite, states that “Moderate- and high-density residential areas are generally situated on fairly flat terrains In urban fire spread, topography plays a secondary role compared with wind dynamics and structure characteristics (Syphard et al. 2017; Masoudvaziri et al. 2021)”. Hence, for these two reasons we consider this outside the scope of our study. Finally, regarding the “location of these structures [and their] proximity to flammable vegetation”, we note that the role of wildland types on home loss has been extensively studied by many of the cited studies in our manuscript.

However, to address your concerns related to proximity of structures to wildland vegetation, wind, and topography we have made the following two revisions. First, we revised the second paragraph in our Introduction to indicate that the above factors have been well studied in the Wildland-Urban Interface fire literature. The revised sentence now

reads, “Indeed, available WUI fire risk and typology literature focuses on wildfire related factors (e.g., wind speeds, topography) as well structure density and proximity to wildland vegetation patches and fuel types. As such, it is often insufficient in capturing the complexity of these disasters in highly urbanized communities that are increasingly at risk of fire.” We also include the following new reference: Balch, J.K., Iglesias, V., Mahood, A.L., Cook, M.C., Amaral, C., DeCastro, A., Leyk, S., McIntosh, T.L., Nagy, R.C., St. Denis, L. and Tuff, T., 2024. The fastest-growing and most destructive fires in the US (2001 to 2020). Science, 386(6720), pp.425-431.”

In addition, this study is focused on fire effects or impacts as we later explain and define to Reviewer 2; not urban fire behavior. So yes, many of the considerations raised by the reviewer would be very interesting for an urban fire behavior study and to parse out the role of wind and other factors. However, for a study of the post-fire impacts wind is less important as we are more interested in the post-fire effects and impacts than how the fire spread during the actual fire event. Further, studies of fire spread in cities are difficult due to issues of scale. Current publicly available active fire data are typically based on VIIRS which has a 375 m resolution, which does not capture the impacts fire spread at the parcel let alone neighborhood-scales presented in this study. The study by Balch et al, 2024 presented later in the study is a good example of the scale of analysis possible with currently available wind, and fire spread data. So, to conclude, as stated in our study aim we explore “...the social-ecological impacts of the Eaton and Palisades fires (2025 L.A. Fires) on communities, neighborhoods, and homes” not the factors behind urban fire behavior and spread. Both types of research are valuable, however the specific goals and approaches are different.

However, to further address this, we now include the above as a limitation. Specifically, the revised last limitation at lines 389-394 now reads, “Finally, our quantitative analyses of the impacts of urban fires did not capture two important groups of factors. Specifically, we did not analyze for the role of environmental factors (e.g., wind speed, topography, fire behavior)²⁰ or proximity of homes to wildland fuel types since these are well studied^{3,6}. Nor do we model for the role of stochastic actions of fire fighters, emergency management professionals, and community members in saving homes during these events.”. We hope this addresses your concerns.

The term "urban fire cycle" as introduced in the conclusion (lines 382–386) as well in the abstract section appears vague and currently lacks sufficient definition in existing fire science literature. While the authors' intention to conceptualize a set of linked processes, i.e., from pre-fire planning through to post-fire recovery is clear, the use of the word "cycle" implies a systematic, perhaps ecological recurrence that may not be appropriate accurate

in 2025 urban fire in LA and overstate the coherence of what appears to be a procedural management sequence.

Thanks for pointing this out. In thinking about this cycle, we are informing the reader that in many studies as well as in applied 'Firewise' community, wildland and WUI fire management, and other emergency management guidelines that we are familiar with, they typically present the planning- mitigation/treatment-response-recovery-rebuilding/rehabilitation steps as a sequential linear process that occurs strictly before and after these events and these all eventually lead to a stable "state" in the system. This is very much akin to the linear succession models that were regularly used in the ecological sciences. For example, see:

- *Ager, Alan A., et al. "Planning for future fire: Scenario analysis of an accelerated fuel reduction plan for the western United States." *Landscape and Urban Planning* 215 (2021): 104212.*
- *Paveglio, Travis B., et al. "Understanding social impact from wildfires: Advancing means for assessment." *International Journal of Wildland Fire* 24.2 (2015): 212-224.*
- *<https://www.poh.usace.army.mil/Missions/Emergency-Response/Hawaii-Wildfires/>*
- *https://www.townofparadise.com/sites/default/files/fileattachments/community/page/42497/eop_2022.pdf*
- *<https://www.fireadaptedwashington.org/6-resources-to-help-communities-recover-after-a-wildfire/>*

So, what we wanted to convey in the Conclusion is that - contrary to the above guidelines and studies- these events appear to be recurring disasters. As seen in the Palisades fire (Fig 1) they are recurring events, but they now appear to occur with greater severity and frequency and as such they are not linear processes that end with a stable end state. But rather, we succinctly posit that these fire-related events and disasters can therefore be approached like cycles akin to the resilience or adaptive cycle(s) that are commonly used in the socio-ecological systems literature when studying complex systems. Unfortunately, delving into this new concept is outside the scope of our study plus we are at limits for our word count. However, we are currently working on a perspective piece on this concept based on experiences with several urban fire events.

But, to address your concern we have revised the last sentence in our abstract by removing this term and we now emphasize the implications of our study's aim. The revised last sentence of our abstract now reads, "Investigating the multiple scales and impacts of wildfire-initiated urban fires provides a valuable next step towards understanding – and developing - policies and actions that mitigate the risk of these future disasters."

Regarding the use of urban fire cycle in our conclusion, we have revised these two sentences and they now state, “As such, adapting the resilience and adaptive cycles that are used in socio-ecological systems literature to understand these new fire regimes in urban ecosystem and communities, could shed light on pre- to post- fire actions. Specifically, thinking of the urban fire cycle as an adaptive process involving pre-fire planning and preparation, firefighting operations and evacuation, and then post-fire response, recovery, and restoration/rebuilding are key and merits future research. We hope this addresses your concerns.

Reviewer #2 (Remarks to the Author):

The following manuscript was reviewed: "Socio-Ecological Impacts of the 2025 Los Angeles Urban Fires on Communities, Neighborhoods, and Homes". The authors explain that their objectives are to look into which socio-ecological factors were associated across the two communities impacted by the 2025 LA County Fires and then determine how the relationships varied based on the scale considered (neighborhood vs. parcel level). The topic covered is of growing importance and works to understand dynamics of who may be most impacted by these wildfire turned urban fire events though the general scope of reference of the results of this work is limited to the two fires analyzed the methods could be used to work to understand other fires and their impacts.

- Line 110 you state that three spatial scales were used for analysis: community, parcel, and neighborhood. I think it would be helpful to give more information on how each of these scales were defined in text even though more detail is given in the supplemental information.

Thank you for the suggestion. To make this point clearer in the main text we moved the first sentence of the Supplementary Materials to the main text as the second sentence of the paragraph highlighted by the reviewer. The text we added is as follows, “Community refers to the Eaton and Palisades areas affected by the 2025 LA Fires. Neighborhood-scale refers to groups of homes located in U.S. Census Blocks, while parcel-scale refers to single structures or homes in individual properties.” We hope this addresses your concerns.

Line 118: for the neighborhood scale analysis you state you used GLMs and robust regression techniques. Can you explain what these robust regression techniques were for those who may not be familiar? Also, please provide reasoning for why these analysis techniques were chosen. What benefit do they provide?

Thank you for your thoughtful feedback and careful review of our manuscript.

The outcome variable, percent destruction, was bounded between 0 and 100%. The dataset contained extreme values that reflected true levels of destruction, not statistical outliers. Diagnostic plots showed that residuals deviated from normality and constant variance, indicating violations of ordinary least squares (OLS) assumptions. To address these issues and confirm the results of the generalized linear models (GLMs), robust regression was used to handle heteroscedasticity and the presence of extreme values. The Robust Regression method in the STATA software, is a statistical technique that reduces the influence of extreme values in the data, providing more reliable parameter estimates when high or low values represent actual observations.

To address this comment, the information from the supplementary section was moved to the main text Methods. Repetitive statements were removed for clarity. The updated version reads as, “The Robust Regression method 43,44 was used to confirm the results of the GLMs due to non-Gaussian error distributions and the potential influence of outliers at the neighborhood-scale.”

Line 119: You switched analysis models for the parcel scale to binary logistic regression. Provide logic as to why you did this? Why did the different scales require different modeling techniques?

We appreciate this comment. At the parcel scale, binary logistic regression was used because the outcome for each structure was binary: destroyed (1) or not destroyed (0). This reflects the actual data collected at the parcel level, where only two possible states are observed. Percent destruction is only defined for aggregated units like neighborhoods and cannot be applied to individual parcels. As such, Logistic regression is the appropriate analytical method for modeling binary outcomes at this scale.

To address this comment, we clarified this point in the main text by explaining that we used binary logistic regression to model a binary outcome (i.e. home destruction) as a function of continuous predictors, given their suitability for non-normal error distributions and categorical responses. The updated main text now reads as “...At the parcel-scale we used Binary Logistic Regression to model a binary outcome (i.e., home destruction) as a function of continuous predictors” in line 180-181. In addition, the text specifies that “ The Binary Logistic regression was used given its suitability for non-normal error distributions and categorical responses” in Lines 184-185.

Line 130-132 I know more detail is provided in the supplemental information, but I think it

would be helpful to state how you did this in the main body of the text so the reader does not constantly have to flip between documents.

We revised this sentence for clarity and highlighted that this was done using Microsoft building footprint data. The revised sentence is as follows, “We quantified the number of nearby structures within Defensible Space Buffer (DSB) zones 0 (0-1.5 m), 1 (1.5-9.1 m), and 2 (9.1-30.5 m), from the main home on each parcel ⁷, and averaged across all parcels in each neighborhood using the Microsoft building footprint ⁴² dataset.” However, we are limited by editorial word counts so there is no way we can include this – and all other reviewer comments- and still make our word count limit. Hence the necessity to include detailed methods in the supplementary section. We hope you understand this requirement.

Line 132: Is structural basal area a term you created to link the urban environment to the wildland since basal area refers to the cross-sectional area of trees at breast height? To me, this just feels like structural density.

Indeed, this is partly the reason we developed this metric- to make these factors more relatable to wildland fire scientists, managers and urban planners. And as far as we know, “structure basal area” hasn’t been used before in the relevant literature. In fact, based on the current literature we reviewed, “structure density” is the sole metric for defining the Wildland Urban Interface (Radeloff et al., 2018). While this definition is useful and similar to tree density or the number of trees per unit area, versus basal area as you explain above in wildland areas, structure basal area provides more information on the footprint or area occupied by the structure across space. It does so because structure density typically means number of structures per unit area (i.e., number of structures ha⁻¹) while we define structure basal area as the footprint area per unit land area (i.e., m² ha⁻¹) similar to tree basal area for wildland areas. The intention is to highlight the importance of the areal extent of individual structures that are potential fuels and the importance or differences in how much surface area is taken up by structures in the different parts of the community compared to the levels of destruction. Accordingly, we have revised the text to highlight the decision to include this metric and how it differs from past literature. The revised text is as follows, “Structure basal area is a novel metric that measure not only density of structures but also their footprint area similar to forest basal area to provide a more nuanced metric of the role of urban density in urban fires.”

Citation:

Radeloff, V. C., Helmers, D. P., Kramer, H. A., Mockrin, M. H., Alexandre, P. M., Bar-Massada, A., ... & Stewart, S. I. (2018). Rapid growth of the US wildland-urban interface raises wildfire risk. Proceedings of the National Academy of Sciences, 115(13), 3314-3319.

Line 175-177 (Figure 2 Caption) I would put the (a) and (b) before the fire you are referencing (e.g. (a) Palisades and (b) Eaton) as that is the typical convention. It took me two read throughs to figure out what your (a) and (b) were referencing at first.

Thank you, this has been revised. The updated figure caption now reads, "Neighborhood-scale fire impacts and % home destruction in the 2025 LA fires for the (a) Palisades and the (a) Eaton communities overlayed with polygons (black) for urban 2020 US Census Blocks for each community and fire history from 1910-2024 (gray)."

Figure 3. Your R2 values are all quite small but most of your p values indicate there is statistical significance to your results. I am just curious as to if you tried any other model types to see if the R2 values improved.

Thank you for this comment. The correlations in Figure 3 represent the relationships between individual neighborhood-level predictors and home destruction (%). We presented these to provide an intuitive representation of the similarities and differences between the relationships in the Eaton and Palisades communities. In our final models we evaluated more than 1,000 model specifications for both Generalized Linear Model and Robust Regression method approaches across both the Eaton and Palisades fires. You are correct that in Figure 3, the correlation for each individual variable was low. When all neighborhood levels were included in a final model for Eaton with $R^2 = 0.53$ (Table 1) and a final model for Palisades with an $R^2 = 0.47$.

Model selection was based on consistency between model results and data visualization, as well as ecological plausibility, and consideration of multiple co-linearities. We assessed correlations among predictors and evaluated multicollinearity using the variance inflation factor (VIF) for each independent variable. All final models included only variables with VIF values below 3 and mean VIF values below 2, which indicates acceptable levels of multicollinearity. This approach ensured that reported results reflected true underlying patterns rather than artifacts of model complexity.

Thank you for this comment. The correlations in Figure 3 represent the relationships between individual neighborhood-level predictors and home destruction (%). We presented these figures to provide an intuitive representation of the similarities and differences between the relationships in the Eaton and Palisades communities.

In our final models we evaluated more than 1,000 model specifications for both Generalized Linear Model and Robust Regression method approaches across both the Eaton and Palisades fires. You are correct that in Figure 3, the correlation for each individual variable was low. When all neighborhood predictors were included in the final models, the overall explanatory power (R^2) improved, resulting in an $R^2 = 0.53$ for Eaton (Table 1) and $R^2 = 0.47$ for palisades.

Also, we based model selection on consistency of model results with socio-ecological theory and plausibility, and consideration of multiple co-linearities. Predictors with a Variance inflation Factor (VIF) greater than 5 were excluded from the final models. This approach ensured that reported results reflected true underlying patterns rather than artifacts of model complexity.

Line 206: this is a line of the main manuscript that separates Figure 4 and its caption

Thank you for catching this mistake. The text spacing has been corrected.

In discussing the impacts of different factors you frequently used the terms significantly but marginally, marginally significant, significant, and strongly but I did not see where these terms were defined. What is the difference between each/what are their cutoffs?

Thanks for the reminder to clarify the cut offs. To clarify our use of this terminology we added the following sentence to at lines 186 to 189 of the Main Text Methods section, “When interpreting our results we used the following definitions for levels of statistical significance: “marginally significant” refers to results with p-values between 0.05 and 0.10, “significant” refers to results with p-values less than 0.05, and “strongly significant” refers to results with p-values less than 0.01.”

These cutoffs were applied consistently throughout the manuscript when describing the strength of associations. We hope these changes make our use of this terminology clearer.

Line 252-254 Can you further explain the property owner race results you were comparing to the census data? Were these self-reported?

Thank you for pointing out the need for clarification. The property owner race was modeled using the “rethnicity” package based on the property owner names reported in the LA County Property Assessor’s data. We revised a relevant sentence in the Methods section at lines 114-116 of the original manuscript to make this clearer. The revised text is as follows, “Multi-scale socio-demographic data was compiled from 2020 US Census data,

US Army Corps of Engineers, and LA County parcel records with the rethnicity model ³⁹ applied to predict race from property owner names.” We also revised the sentence referenced by the reviewer to highlight that the property owner race data were modeled. The revised sentence is as follows, “Modeled property owner race results are in-line with US Census data for Eaton, however results for Palisades were not consistent (Figure S2), so modeled property owner race data were not used in subsequent analyses due to concerns about model accuracy for Palisades.” We hope these revisions address your concerns.

Line 324 "the most affected, higher density southeastern areas of the community" I was confused by what this referred to. Are you saying that newer homes tended to be in areas with greater impacts from these fires (e.g. the edges of communities?)

Thank you for pointing out the need for clarification. We have revised the sentence to highlight how urban development patterns in Palisades appear to differ from the typical pattern of urban growth at community edges near wildland vegetation. The revised text is as follows, “the most affected (Figure 2a), high density southeastern areas of the community (Figure S4a) instead of the community edge adjacent to wildland vegetation.” We hope this revision addresses your concerns.

Line 325-326 Not all WUI fire literature report that lower housing densities are associated with greater fire risk. For example, I know off the top of my head that Caggiano et al (2020) reported that intermix WUI areas experienced the highest rates of housing loss to wildfire versus interface and that building loss mostly occurs in areas with low building density and high vegetation cover.

Thank you for pointing this out and the need for clarification. Accordingly, we revised the text to highlight the differences in where homes were lost between the 2025 LA Fire events and the existing WUI fire literature. We also added a citation to the Caggiano (2020) study mentioned by the reviewer. The revised text is as follows, “These neighborhood-level findings of high home losses in dense urban communities, contrast with the WUI fire literature that reports that lower housing density (i.e., urban periphery, WUI) is associated with greater fire risk ^{4,51}.” We hope this revision addresses your concerns.

I like that the authors clearly stated the limitations of their analysis (lines 340-353)

Thank you.

In general, I think the work presented in this paper is needed and a step forward in

understanding how fires impact communities; however, I personally would like more detail to be provided in the methodology. With the current level of detail provided, I am unsure I would be able to properly replicate the processes were I to try.....I think it would be helpful to incorporate more of the supplemental information into the main body of text.

Thank you for this suggestion regarding the repeatability of our methods. We agree that the original version of this manuscript lacked sufficient detail in the Main Text to allow for the results to be properly replicated. In fact in the original working draft all methods were included in the main text, but then we ran into the word count limits for this journal. But, as suggested by the reviewer and described in detail in our previous responses to reviewer comments, we have revised the Methods section in our Main Text document to include more of the details originally provided in the Supplementary Materials. Furthermore, we have revised both the Main Text and Supplementary Materials for clarity to ensure readers are able to replicate our results. We have also added a link to a FigShare repository which includes the data and code used in the analysis to further add to the reproducibility of the study. When considering these revisions – plus word count limits - in the Main Text and Supplementary Materials we believe that the current draft provides enough details to allow readers to reproduce our results. We hope these changes address your concerns.

Additionally, I think the results could focus more on the story they are telling. I think more of the why are the findings important should be included when discussing the analysis results.

Again, thank you for this comment. Overall, we feel that in large part the revisions we did in response to your previous comments plus the following revisions addressing comments from Reviewer 2 have now, hopefully, better explained the importance of our findings. However, as requested we edited and revised text in both our Results and Discussion section.

Specifically, in our Results section, we have revised and edited text when necessary, to clarify specific findings. For example, we have replaced “impacts” with either “home destruction” or “fire exposure” throughout the Results and Discussion sections to be more effective in communicating why this finding is important. We have also edited sentences in the first paragraph to better communicate findings related to the temporality and influence of historic fire exposure events at the community level. We have also revised table titles to explain all symbols and acronyms and have named all analyzed variables consistently.

Then, in our Discussion section we now also emphasize and explain how our development and use of urban density metrics (i.e. structure basal area, number of homes

in Defensible Space Buffers) is an additional contribution of this study. And we also now mention how such metrics can be used for other purposes. Specifically in the last paragraph we have added new text that states “Indeed, some socio-ecological, parcel scale variables used in our analyses (e.g., structure basal area, number of homes in DSBs, urban tree cover) could also be used for developing or improving urban fire behavior models.” We hope these and the other response address your concern.

Reviewer #2 (Remarks on code availability):

I did not see that any code was provided.

Thank you for noting this oversight. We have included a link to a FigShare (and CodeOcean) repository as well as a Data Availability Statement to the current draft. The links to the FigShare and CodeOcean repositories will be made public once the manuscript is accepted.

Reviewer #3 (Remarks to the Author):

The manuscript “Socio-Ecological Impacts of the 2025 Los Angeles Urban Fires on Communities, Neighborhoods, and Homes” provides a detailed spatial and statistical analysis of the destruction caused by wildfire-initiated urban fires in two contrasting communities in Los Angeles.

Comments:

Can the authors please clarify the concept of “wildfire-initiated urban fires”? I assume that the authors mean wildfires that affect, reach or enter urban areas/cities, but this is not clear. I understand your point-of-view, expressed at the end of page 3 and beginning of page 4, but I believe that “wildfire-initiated urban fires” is a confusing terminology. On the other hand, “urban fires” are mentioned in the title and “wildfire” is one of the keywords. An urban fire is usually associated with a human trigger/cause/fault (e.g. short circuit), having no connection to natural causes, especially to wildfires. So, there are different mechanisms and rational intervening here. At the same time, the authors mention “wildfires”, but writes the following: “As such, events like the 2025 Eaton and Palisades fires in L.A. have been deemed wildfire initiated urban fires; not wildfires”. In sum, the authors need to clarify these terms and their correct use.

Thank you for pointing out the need to clarify terminology. We have updated references to “wildfire-initiated urban fires” to “urban fires initiated by wildfires” throughout

the manuscript. We believe this updated terminology more clearly describes the theoretical framing of the manuscript and also aligns with the presentation of the concept of urban fires per Calkin et al, (2023). For example, the sentence mentioned by the reviewer has been revised as follows for clarity, “As such, events like the 2025 Eaton and Palisades fires in L.A. have been deemed urban fires initiated by wildfires ²¹.” We hope these revisions have addressed your concerns.

Citation:

Calkin, D. E., Barrett, K., Cohen, J. D., Finney, M. A., Pyne, S. J., & Quarles, S. L. (2023). Wildland-urban fire disasters aren’t actually a wildfire problem. Proceedings of the National Academy of Sciences, 120(51), e2315797120.

- Following the previous comment, what are the differences in fire behavior in this event (caused by a wildfire), compared to an event caused by a short circuit?

This is a good question and is likely not fully understood. At least for the Eaton fire there did appear to be a link between electric infrastructure, high winds, wildland vegetation and urban structures. We would hypothesize that a similar fire could have occurred without wildland vegetation being involved if urban vegetation or homes had been ignited instead of wildland vegetation by the power lines. For the Palisades Fire the initial ignition, appears to have resulted from a flare up of an arson-caused fire that occurred one week prior.

Page 4, lines 70-72: “Key to this is who lived in these communities and what types of homes, neighborhoods, and environmental amenities were lost or damaged as these factors”. This configures an exposure and vulnerability study. However, these terms are rarely used in this manuscript, while these should be a focal point of the study.

Thank you for this comment and suggestion. In this revised version we now highlight – and define- the “exposure” and fire vulnerability implications of our study in the Introduction, Discussion and Conclusion sections. In our Introduction we now include the following revised sentences in our Aims and Objectives paragraph as well as in the Methods-Direct Fire Impacts section:

- *“Key to this is who lived in these communities and what their exposure, adaptive capacity, and vulnerabilities to urban fires are²⁷, as well as what types of homes, neighborhoods, and environmental amenities were lost or damaged as these factors – and scales- are rarely addressed in the above literature in an integrated manner*

^{27,30} . “

- *“Scale matters in analyses of community-specific contexts, fire vulnerability, and potential impacts as they are highly relevant in mitigating and responding to these events²⁹.”*
- *“We use exposure hereafter to refer to homes and structures that were inside the fire perimeters but that were not specifically damaged or destroyed.”*

Second, for in our Discussion section we have revised the following 3 sentences in the first and last paragraphs:

- *“...our results integrate urban morphology, socio-demographic, and fire impact data to better understand the multi-scale socio-ecological complexity ofcommunity exposure and vulnerability to urban fires^{18,27}.... and improving other fire-related socio-ecological assessments^{45,52}.”*
- *“..., future research on wildfire initiated urban fires can build off our approach and methods to better understand ... impacts on highly populated urban communities and their vulnerabilities...”*
- *“...This is particularly relevant ... since the parcel scale will determine the vulnerability of individuals and families, tenure, human actions....”*

Finally for the Conclusion, we have modified the following 2 sentences:

- *“This has implications for defining and mapping “vulnerable communities at risk of fire”.*
- *“Such knowledge would provide a valuable next step towards understanding not only what happens during urban fires and who is impacted and vulnerable, but also how and what policies...”*

The abstract lacks information regarding the data and methods used in the manuscript. This should be added to the abstract. Additionally, the results mentioned in the abstract are a little vague. It is necessary to improve these points.

Thank you for pointing out these areas for improvement. We revised the abstract sentence describing methods to add a bit more detail to the data and methods used. The revised text is as follows, “Geospatial analyses and econometric modelling explored the relationships between urban morphology (e.g., pre-fire tree cover, urban density metrics, etc.), socio-demographic (e.g., per capita income, education level, racial diversity, etc.) factors and home destruction at both the neighborhood- and parcel-scales.” We also added a sentence to the abstract results to provide more specifics about the strength of the relationships in the models. The added sentence is as follows, “At the neighborhood-scale, socio-ecological factors were more highly correlated in Eaton ($R^2 = 0.53$ versus $R^2 = 0.47$)

while at the parcel-scale, fire impacts were better explained by socio-ecological factors in Palisades (pseudo- $R^2 = 0.10$ versus pseudo- $R^2 = 0.03$).” We hope these changes address your concerns.

The study area section must include a map representing the location of the studied communities/neighborhoods. Information regarding the size/area of Eaton and Palisades should be included in this section. The authors should also include more information regarding: 1) the meteorological conditions before and during the event; 2) the rainfall amounts during 2024 (spring, summer and autumn months). Lastly, what was the proportion of buildings affected by fires during this event?

Thank you for the suggestions. We revised Figure 1 to include a map of the locations of the studied neighborhoods and communities as a panel. We also revised the text of the study area conditions to highlight the metrological conditions and rainfall amounts during 2024. We revised the text to more specifically describe the precipitation, drought, and meteorological conditions and cited a detailed explanation produced by UCLA scientists in mid-January 2025 (Madakumbura et al, 2025). The neighborhood-scale proportion of building impacted during the event is reported in Tables S3 as well as paragraph 3 of the Results. The revised sentence referencing meteorological conditions is as follows, “The region’s recent climate history, marked by wet winters in 2022-2023 and 2023-24 (double the 1877-2024 mean) followed by hot and dry conditions from April 2024-January 2025 coupled with an extreme wind event on January 7th and 8th 32, created ideal conditions for fire ignition and spread 33.” We hope this addresses your concerns.

Citation:

Madakumbura, G. et al. Climate Change A Factor In Unprecedented LA Fires. Sustainable LA <https://sustainablela.ucla.edu/2025lawildfires> (2025).

- In data and methods section, the authors mentioned “robust regression techniques”. This should be properly explained.

Thank you for the feedback. We missed this explanation in our main text previously. The outcome variable, percent destruction, was bounded between 0 and 100%. The dataset contained extreme values that reflected true levels of destruction, not statistical outliers. Diagnostic plots showed that residuals deviated from normality and constant variance, indicating violations of ordinary least squares (OLS) assumptions. To address these issues, a Robust Regression method was used to address issues of heteroscedasticity and the presence of extreme values. This Robust regression method is a statistical technique that reduces the influence of extreme values in the data, providing

more reliable parameter estimates when high or low values represent actual observations. In addition, generalized linear models (GLMs) were applied with appropriate error distributions and link functions to properly model the bounded nature of the percentage data.

Thank you so much for the feedback. We missed this explanation in our main text previously. The outcome variable, percent destruction, was bounded between 0 and 100%. The dataset contained extreme values that reflected true levels of destruction, not statistical outliers. Diagnostic plots showed that residuals deviated from normality and constant variance, indicating violations of ordinary least squares (OLS) assumptions. To address these issues, robust regression was used to handle heteroscedasticity and the presence of extreme values. The Robust Regression method in the STATA software is a statistical technique that reduces the influence of extreme values in the data, providing more reliable parameter estimates when high or low values represent actual observations. In addition, generalized linear models (GLMs; Gaussian Family Identity Link) were applied to confirm the consistency of the results.

To address this comment, the information from the supplementary section was moved to the main text of the Methods. Repetitive statements were removed for clarity. The updated version reads as “The Robust Regression method^{43,44} was used to confirm the results of the GLMs due to non-Gaussian error distributions and the potential influence of outliers at the neighborhood-scale”

- Have the highly correlated independent variables been eliminated?

Thank you for the question. Yes, highly correlated independent variables were eliminated from all models. We assessed correlations among predictors and evaluated multicollinearity using the variance inflation factor (VIF) for each independent variable. All final models included only variables with VIF values below 5, which indicates acceptable levels of multicollinearity.

- At the beginning of page 7, this is mentioned: “the proportion of homes destroyed by fire in each neighborhood and the number of homes destroyed in each parcel”, but it is not clear which is/are the dependent variables used.

Thank you for highlighting the need for clarification. To improve clarity we revised text and we split the sentence into two shorter sentence and edited the wording. The revised text is as follows, “We analyzed the degree of fire impacts to individual homes at the neighborhood and parcel level by using CALFIRE’s Damage Inspection (DINS) data⁴⁴. We

quantified the proportion of homes destroyed at the neighborhood-scale and the number of homes destroyed at the parcel-scale.” We hope these revisions addressed your concerns.

- The methods used in this study should be better explained. I suggest that the authors build a flowchart with the steps followed.

Thank you for pointing out that the methods used in the study need to be better explained. In response to comments from all three Reviewers we made major revisions to the Methods section to improve clarity and moved text from the Supplementary Materials section to make the Main Text Methods clearer. Those revisions are explained in detail in our responses to comments from Reviewers #1 and #2. Given the previous revisions we did in response to Reviewer 1 and 2’s requests that we include specific methods from our Supplementary Materials section in the main text, to improve repeatability, and other revisions to improve the clarity of the Methods we feel that we have already addressed this comment. Please refer to our responses to Reviewer 1’s comments about Table S1 and our use of 2023 American Community Survey and Reviewer 2’s comments at line 130-132 about the study area and at line 252-254 about the predicted property owner race results. We hope this addresses your concerns regarding our methods.

- Figure S1 should be included in the text. This is important to understand the differences between the two study areas.

Thank you for the suggestion. We agree that Figure S1 provides valuable information to describe the differences between the two study areas. However, Figure S1 is currently cited only twice throughout the manuscript. For that reason and due to word count limits, space we believe that information presented in the text is sufficient to describe the differences between the two study areas. However, we have revised and added a map (Figure 1a) to better demonstrate the differences between the two study areas and their geographic location- please refer to the following comment. We hope this addresses your concerns.

- Figure 1 leaves much to be desired. The lines are too thick, and it is unclear what values are reached in 2025.

Thank you for the suggestions. As requested, the lines in Figure 1 have been made thinner, the trends are shown in bars, and the figure has been widened slightly to make the location of the lines in 2025 more visible. We have also added a study area map and the location of urban and non-urban fire affected areas. We hope this addresses your concerns.

- Page 7, lines 159-160: “About 30% of Palisades was impacted by historic fires from 1910-2024, while fires in 1938 and 1993 alone impacted >25% of the community (Figure 1)”. This sentence is confusing. The authors mentioned 30% between 1910 and 2024. This value came from where? It is impossible that this the average value in this period. On the other hand, this chart somehow contradicts past sentences (e.g. “Such urban conflagrations were historically common in cities prior to the 1900s or during periods of armed conflict”). The high peak in the chart show that these events were not uncommon and new in Palisades (despite the extreme event of 2025).

Thank you for pointing out the need for improved clarity. As you point out several of the metrics in this sentence needed to be updated to align with Figure 1. We have revised the sentence accordingly, “In 1938 as much as 39% of the community of Palisades was impacted by a fire, and fires in 1978 and 1993 impacted 19% and 21% of its area, respectively (Figure 1b). .” We have also added a new final sentence to the paragraph to highlight the contrast with the 2025 fires, “In contrast, the 2025 fires impacted 57% of Palisades and 53% of the Eaton communities (Figure 1).” We hope this addresses your concerns.

Additionally, as noted by the reviewer, it is true that a relatively large portion of the Palisades community was impacted by fires before 2024. However, as seen in Figure 2, prior to 2025 these fire events occurred mostly at margins of these communities that are non-urban areas adjacent to wildland areas or what is traditionally considered the “Wildland Urban Interface”; not “urban” as defined in our study. As stated in our introduction, the unique aspect of the 2025 fires was that neighborhoods that are clearly urban, and not adjacent to wildland areas, were heavily impacted. Therefore, this does not contract the point that, especially for the Eaton fire, a much larger portion of urban areas and neighborhoods, were impacted by the 2025 fires.

- Figure 2 needs to be improved. The lines of the polygons are too thick (red and blue). There many areas covered in black. This means that there were no houses destroyed, despite these were affected by fires? There are some isolated areas/polygons with no connection with the other polygons. The readers will not understand what happens between these polygons/areas. Forests? What is the relationship between the proportion of houses destroyed and the distance to the polygons’ edges? This should be quantified.

Thank you for these suggestions. To improve the figure, we have revised the figure to remove the 2025 Fire Perimeter as the map location is already filtered to include only US Census Blocks that were within the fire perimeter. We have also changed the labeling (now

“Fire History”) and color of the polygon, added shading to the polygon to clarify which areas had fire history from 2010-2024. This polygon is a combination of all fire perimeters from 1910-2024. We hope these changes address your concerns.

- Figure 3 only presents some independent variables and mixes urban morphology and sociodemographic characteristics. Why did the authors choose these 6 variables? An explanation is needed in the text.

Thank you for pointing out the need for further explanation. The prior text could have more clearly referenced the six panels from Figure 5. But to respond to your comment, we selected six representative variables that have previously been reported in other studies to illustrate the range of comparisons between variables in the two communities at the neighborhood level for both sociodemographic and urban morphology variables as well as based on correlation with home destruction based on Figure 4.

We chose DSB 0 and structure basal area to demonstrate similarities. We chose English speakers (%), African American (%), Urban Tree Cover (%), and Bachelor’s Degree (%) to illustrate variables where the relationships differed across the two communities. The other variables used in both study areas are shown in Figure S5. So, to address your comment, we have added the following sentence to the second paragraph of our Results, “Figure 3 displays some representative urban morphology and sociodemographic variables that have previously been reported in relevant literature^{1,8,41}.”. Also, we added another sentence to better describe the socio-demographic variables (Figure 3 d, e, f). The text is as follows, “In contrast, six out of the fourteen neighborhood-scale socio-demographic variables showed divergent relationships (Figure 3d, e, f; Figure 4; Figure S5).”

- Page 9, line 181: “exposure” should be replaced by affected. If I understand correctly, these houses were affected/hit by previous fire events. So, it is more correct the use of affected.

Correct “exposure” means that the homes were located inside fire perimeters. Furthermore, for past or pre-2025 fires we do not know if the homes were “destroyed” or “survived” because we do not have the Damage Inspection (DINS) data for the period. However, we do have the DINS data for 2025 fires and thus we know which homes were destroyed and which survived. That said, we believe our interchangeable use of “Impacts, “exposure”, “affected” and “Destroyed/destruction” adds to this confusion.

So to address your comment, we now add text in our Methods- Direct Fire impacts paragraph (last sentence) that states, “We use exposure hereafter to refer to homes and structures that were inside the fire perimeters but that were not specifically damaged or destroyed.” Otherwise we specify when homes were “destroyed” or “exposed” in our methods and results section. But then in our Discussion and Conclusion we refer to “impacts” as the sum of exposed and destroyed homes. Specifically in the fourth sentence of our first paragraph we revised the sentence to explain this, and it states, “Conversely, our results integrate urban morphology metrics, as well as socio-demographic, and fire impact data (i.e., exposure to fire, home destruction) to better understand the multi-scale socio-ecological complexity of these events and their role in...”

- Figure 4 should present the correlation coefficients of all variables considered in both study areas.

Thank you for the suggestion. As suggested by the reviewer, we created a new version of Figure 4 to better show the correlation coefficients of all the variables considered in both study areas (Figure R1). However, as shown below, the Figure was too busy and difficult to interpret due to displaying too much non-critical information. Therefore, we will respectfully disagree with you and keep the original figure which specifies and focuses the reader on those variables that present across both scales (neighborhoods and parcel) to make the comparison. We hope this is acceptable to you.

Figure R1. Correlations of socio-ecological characteristics with home destruction at the neighborhood and parcel scales. Blank panels represent socio-ecological characteristics that are not available at either the parcel or neighborhood scale.

AUTHOR'S RESPONSE TO REVIEWERS

REVIEWER COMMENTS

Reviewer #2 (Remarks to the Author):

The following manuscript was reviewed for the second round of peer review: "Socio-Ecological Impacts of the 2025 Los Angeles Urban Fires on Communities, Neighborhoods, and Homes". In general, I think the manuscript has been improved; however, in giving the paper a read through I still think there are some general clarity and proof-reading issues. I believe the topic covered is of growing importance and works to understand dynamics of who may be most impacted by these wildfire turned urban fire events though the general scope of reference of the results of this work is limited to the two fires analyzed the methods could be used to work to understand other fires and their impacts. The changes to the manuscript did make the methods more clear and replicable and less reliant on the supplemental information. Below my direct comments are outlined:

- Line 33: Money formatting is non-traditional. I would suggest using \$76-131 billion USD.

Thanks for pointing this out. We have revised our monetary units to the suggested formatting.

- Line 37: colon is not needed

We have removed the colon as suggested.

- Line 39-45: Why are you putting the terms communities and structures in ""? This is not done elsewhere in the paper.

We have removed the quotation marks here for clarity and consistency with the rest of the manuscript.

- Line 61-66: This sentence is very long and confusing. I had to read it multiple times before I understood what you were saying. Consider splitting it up into more than one sentence for clarity.

We see your point, so we have revised this statement as two sentences as suggested by the reviewer. The revised text is as follows, "More recently these urban fires are bringing to light the importance of urban fuels (e.g., building types, construction materials, and ornamental vegetation²⁵) in influencing fire behavior as opposed to wildland fires that are driven by natural vegetation. Urban fires also result in greater loss of life, infrastructure and property as well as in indirect impacts such as human displacement and human health in distant populations due to

emerging air, water and soil pollutants (e.g., Lithium from auto batteries, toxic compounds in water systems, soil lead pollution^{26,27}).

- Line 81: em-dash around "and scales" is not needed

We have removed the em-dash as suggested.

-Line 103-105: Once again non-traditional money formatting and estimated used twice. Consider rewording to reduce repetition.

As addressed earlier, we have revised the formatting as suggested by the reviewer.

- Line 106-108: Common separated list of areas was confusing. Consider doing:

"...including Altadena, Pasadena, and Sierra Madre area (Eaton hereafter); Pacific Palisades (Palisades hereafter); and adjacent wildland areas in the San Gabriel and Santa Monica Mountains." or something similar.

We have revised as suggested by the reviewer. The new text is as follows, "These fires primarily impacted urban areas and communities, including Altadena, Pasadena, Sierra Madre area (Eaton hereafter); Pacific Palisades (Palisades hereafter); and adjacent wildland areas in the San Gabriel and Santa Monica Mountains.

- Line 158-160: Glad you included more information on what structural basal area is BUT I would like to see WHY this metric is better than structure density which is what is commonly accepted and used in literature. What benefit does this new metric provide?

*Thanks for pointing this out and for the opportunity to better justify this metric. In thinking about how we called this metric, we now realize that the use of "basal" and steering the reviewer and potential readers to the forestry literature as its basis, is causing confusion. However, "building footprint area" is regularly used in the urban planning, architecture, and geography literature (See: Durst, Noah J., et al. "Building footprint-derived landscape metrics for the identification of informal subdivisions and manufactured home communities: A pilot application in Hidalgo County, Texas." *Land Use Policy* 101 (2021): 105158; Heris, Mehdi P., et al. "A rasterized building footprint dataset for the United States." *Scientific data* 7.1 (2020): 207). Therefore we will now refer to "structure basal area" as "building footprint area" in this revised version of our manuscript to avoid confusing readers.*

*Regarding the benefits of this metric, fire management organizations in California (e.g. US Forest Service, Cal Fire) and elsewhere use "structure/housing density", defined as the number of housing units per hectare, as the metric for identifying what a community or human settlement is or is not and its exposure to wildfire (Evers, Cody R., et al. "Archetypes of community wildfire exposure from national forests of the western US." *Landscape and urban planning* 182 (2019): 55-66). Similarly, the "number of houses/structures per unit area", and their distance to contiguous patches of natural vegetation, is the typical approach to defining and*

mapping the Wildland-Urban Interface and its implication for fire managed and planning (Radeloff et al., 2018).

Although this number of structures per unit area metric can suffice for analyzing wildland fires, it is not sufficient in, 1) differentiating the types of structures (i.e., single family home, multifamily home, department store, mobile home, barn, shed or recreation vehicle) and 2) the space these structures occupy per unit area, in urban areas. As such, there is a need to account for other aspects of housing density such as structure separation distances and proximity to adjacent structures since these have implications for direct flame contact and radiant heat during urban fires; hence our use of “building footprint area” (<https://www.nist.gov/el/fire-research-division-73300/wildland-urban-interface-fire-73305/hazard-mitigation-methodology-9>). Additionally, this building footprint (or floor) area metric has a long use in urban planning in characterizing the spatial structure of cities and data are readily available for use in future studies (See Heris, Mehdi P., et al. "A rasterized building footprint dataset for the United States." Scientific data 7.1 (2020): 207, and citation therein).

So, to address your request, we now revise this text and specify why we propose this metric with the following revised text. “Structure footprint area is a metric that we propose to measure not only density of structures (i.e., number of structure per unit area) as is done in wildfire exposure and WUI studies^{7,21}, but also to differentiate the size and space these structures occupy per unit area as is done in the urban planning and geography literature^{44,45}.

- Line 171-172: How did you determine which structures were "major" and which were miscellaneous? By area?

We used major structures as defined by the CAL FIRE in the DINS data. We have revised the sentence to clarify this point. The revised sentence is as follows, “When considering fire impacts we only considered impacts to major structures (e.g., homes) as observed by the CALFIRE DINS data collectors and excluded parcels that had only miscellaneous structures (e.g., garages) from the data set.”

- Line 195: 100+ years? This is confusing. Should it just be 100 years?

We have revised “100+” to “100” as suggested by the reviewer.

- Line 201-202: Why were these chosen for analysis? I want more information that they have also be reported in other literature.

Thank you for this suggestion. But first and just to clarify, these 6 were not the only variables selected for our analysis. Tables S1 and S2 display the over 30 community, neighborhood, and parcel scale variables that were analyzed in our study. In lines 202- 206 we refer the reader to other figures where some of these variables were analyzed, and findings displayed in the form of figures. Clarifying this is important since displaying all these 30+ figures would be too much. So, as stated in lines 201-202, “Figure 3 displays some representative urban morphology and

sociodemographic variables”. We deliberated on which of these 30+ variables to display and at what scale. We decided that these 6 neighborhood scale variables were the most informative and relevant to urban and Wildland-Urban Interface communities and that these 3 references would suffice in justifying their use.

However, as requested, we would like to provide you with a more detailed explanation. Specifically, the 6 representative variables shown in this Figure were selected to better represent neighborhood scale socio-ecological factors that have been previously documented as influencing wildfire: exposure, effects, damage and recovery outcomes. For example, the upper biophysical or built-environment factors we present are widely cited correlates for characterizing structural fuel loads and drivers of structure-to-structure fire spread in wildland-urban interface and urban settings. These include housing density/ structure footprint area (formerly structure basal area) and proximity to other structures (i.e., Defensible Space Buffers). Similarly, vegetation cover, usually expressed as tree cover, has also been used as a biophysical correlate of home damage. These factors have been documented by Syphard et al., (2012). Aguirre et al. (2024), Knapp et al (2021) and Carlson et al. (2022).

That said, we do admit that the “number of structures within defensible space buffers (DSB)” variable was not justified or explained. So, similar to our structure footprint area (formerly structure basal area) variable, we propose that the number of structures in DSB 0 (number of structures within 2 meters of a home) is a new, and policy, relevant housing density metric that can be used to better characterize urban fires (50, Escobedo, et al. "Exploring urban vegetation type and defensible space's role in building loss during wildfire-driven events in California." *Landscape and Urban Planning* 262 (2025): 105421). Indeed, recent policies in the study area around Defensible Space Buffers in California cities have been controversial as they lack evidence regarding the effectiveness of these buffers in home survival during urban fire events - hence our proposed use of this metric. Please see these newspaper articles for background on this problem:

- <https://www.latimes.com/environment/story/2025-01-17/lessons-from-the-burn-zone-why-some-homes-survived-the-l-a-wildfires>
- <https://www.sfchronicle.com/projects/2025/california-fire-zone-zero-rules/>
- <https://www.latimes.com/lifestyle/story/2025-09-15/proposed-zone-zero-rules-would-remove-all-plants-within-five-feet-homes-fire-areas>

With regard to the 3 socio-demographic variables: educational attainment (% Bachelor's degree), language isolation (Non-English speakers), and racial composition (% African American), these are commonly used indicators of social vulnerability and adaptive capacity in disaster and wildfire research (e.g., Yadav et al., 2024, Hossain and Smirnov 2023; Palaologou et al., 2019, and Jennings 2013,). These 3 variables have also recently been shown to be emerging indicators of vulnerability to urban fires, redlining and housing conditions/ tenure in high fire risk

areas, and access to fire mitigation resources in urban southern California (Thomas et al., 2022 and Yadav et al., 2024).

So, to address your concern we have revised this text and added a new citation (Escobedo et al., 2025) to better justify and clarify that the variables shown are consistent with prior studies and that two of these are new metrics we propose. The revised text now states, “Figure 3 displays some representative urban morphology^{4,8,9,52} and sociodemographic variables^{1,39,47,48} that have previously been reported as influential correlates of home damage in urban and WUI fires. Additionally, we propose ‘structure footprint area’ and ‘number of structures in Defensible Buffer Space 0’ as complementary metrics for better understanding urban fires^{53,54}.”

- In the discussion I would like to see more of the story being told. Why are these findings significant and what do we gain from them? This was hit on some but I feel like more could be done. I felt that the claims regarding the importance of urban morphology over vegetation, scale-dependent relationships, and sociodemographic disparities in impacts were well supported, but I'd love to see more information and discussion regarding the mechanism behind opposing UTC relationships between communities and how the omission of environmental factors (wind, topography, fire behavior, etc.) could impact results.

Thank you for these suggestions. We have revised this section to better interpret the Urban Tree Cover (UTC) findings and revised our limitations section to address the impacts of excluding the environmental factors you mention. Specifically, we added several sentences and a new reference to describe potential differences and hopefully better explain our findings between UTC and home damage for Palisades and Eaton. The revised text is at lines 346-351 and reads as follows, “The differences in the relationship between pre-fire UTC and home destruction in the two communities appear robust when collinearities are considered as they remain when included in multiple regression models (Tables 1, 2, 3, 4). However, the differences could be due to differences in tree structure and composition (e.g., tree heights, density, species) or how trees are maintained (e.g., irrigation, pruning) across each community⁵⁴. Similarly, differing firefighter and homeowner actions and neighborhood scale urban morphology interactions could be factors⁵⁴. Regardless, a more detailed forensic analysis of specific individual parcels and homes would be required to untangle these potential drivers.”

We also revised our study limitations sections to better address the impact of not including the environmental factors you mention. Specifically we believe that including them would not have changed our results because as we highlighted in our previous response to reviewers, our models are meant to be descriptive – not predictive – and meant to better understand which socio-demographics and neighborhoods/ homes in Eaton and Palisades were most impacted. To make this point clear to the reader we also added a concluding sentence highlighting how environmental factors along with the urban morphology and socio-demographic

factors we studied, could indeed be used in the future development of predictive models of urban fire behavior and/or impacts. The revised text is at lines 396-402 and reads as follows, “Specifically, we did not analyze for the role of environmental factors (e.g., wind speed, topography)²⁰ or proximity of homes to wildland fuel types since these are well studied^{3,6} and likely would not have changed our results given the descriptive nature of our analysis. Nor did we model for the role of stochastic actions of fire fighters, emergency management professionals, and community members in saving homes during these events. But we acknowledge that environmental factors, parcel-scale fire suppression actions, urban morphology, and socio-demographic could all be important factors in developing future predictive models of urban fire behavior and impacts.”

- Line 420: Include a citation for "Urban fire disasters are occurring more frequently and globally"

Done, we have revised this sentence to better cite and justify this statement. “Urban fire disasters are occurring more frequently, not only in western North America¹, but globally⁶⁸ across various biomes and context as well^{2#}”

- Line 427-429: This feels a little vague for conclusions. What specifically could use further study and why? How can it build upon your work?

We revised the sentence to be more specific about the need for parcel level analysis of socio-demographic variables which are typically available at the neighborhood level from the US Census. The revised text is as follows, “Our findings and analyses of the impacts of the 2025 L.A. urban fires span multiple scales, however due to the limitations of U.S. Census Data, socio-demographic impacts (e.g., age, race, ethnicity, income, etc.) at the scale of individual parcels are little studied and need to be understood further.”

- Line 433-438: How does this connect to your findings?

We revised the first and last sentences of this paragraph to better connect this information to our findings and so that it is clearer to the reader. The revised text of the paragraph is as follows with the middle part of the paragraph omitted, “In conclusion, based on our findings that highlight the importance of scale of analysis and accounting for both biophysical and socioeconomic drivers of urban fire impacts, adopting the resilience and adaptive cycles from the socio-ecological systems literature to understand these new fire regimes in urban ecosystem and communities could shed light on pre- to post- fire actions. ... For urban fires that behave similarly to the 2025 L.A. fires, considerations across various spatial scales provide a valuable next step towards understanding not only what happens during urban fires and who is impacted and vulnerable, but also how and what policies and

management actions can be formulated and successfully implemented to mitigate the risk of future disasters.”

- In conclusions would be good to synthesize key finds. The conclusions covers novelty of approach before jumping to future frameworks without reminding us of what your knowledge your study created.

As addressed in previous responses, we have revised both paragraphs in this section to better link our findings with the knowledge implications of our conclusion as suggested by the reviewer.

- Line 431: Typo. "paradigm shits" should be "paradigm shifts"

Apologies, thank you for catching the typo! This has now been corrected.

Reviewer #3 (Remarks to the Author):

After the changes made by the authors, the manuscript can be accepted in its present form.

Thank you for taking the time to review our revised manuscript!